# BiGR: Harnessing Binary Latent Codes for Image Generation and Improved Visual Representation Capabilities

**Shaozhe Hao**[1]    **Xuantong Liu**[2]    **Xianbiao Qi**[3*]    **Shihao Zhao**[1]    **Bojia Zi**[4]
**Rong Xiao**[3]    **Kai Han**[1†]    **Kwan-Yee K. Wong**[1†]
[1]The University of Hong Kong    [2]Hong Kong University of Science and Technology
[3]Intellifusion    [4]The Chinese University of Hong Kong
{szhao,shzhao,kykwong}@cs.hku.hk qixianbiao@gmail.com kaihanx@hku.hk
Project page: https://haoosz.github.io/BiGR
Code and models: https://github.com/haoosz/BiGR

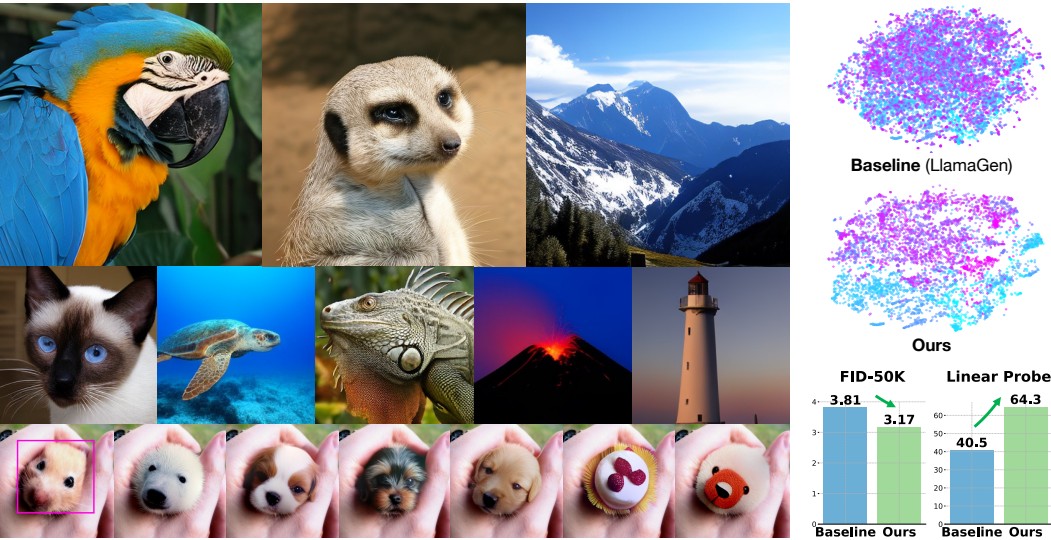

Figure 1: **BiGR generates high-quality images while improving the discriminative capabilities of the representations.** **Left**: Generated 512×512 samples, 256×256 samples, and class-conditional editing samples. **Right**: BiGR *vs.* LlamaGen (Sun et al., 2024). We visualize image features from 100 classes in ImageNet-1K validation split using t-SNE (van der Maaten & Hinton, 2008), where the same color indicates the same class. Our model produces features with greater discriminative separability and enhances both generative and discriminative performance.

## Abstract

We introduce BiGR, a novel conditional image generation model using compact binary latent codes for generative training, focusing on enhancing both generation and representation capabilities. BiGR is the first conditional generative model that unifies generation and discrimination within the same framework. BiGR features a binary tokenizer, a masked modeling mechanism, and a binary transcoder for binary code prediction. Additionally, we introduce a novel entropy-ordered sampling method to enable efficient image generation. Extensive experiments validate BiGR's superior performance in generation quality, as measured by FID-50k, and representation capabilities, as evidenced by linear-probe accuracy. Moreover, BiGR showcases zero-shot generalization across various vision tasks, enabling applications such as image inpainting, outpainting, editing, interpolation, and enrichment, without the need for structural modifications. Our findings suggest that BiGR unifies generative and discriminative tasks effectively, paving the way for further advancements in the field. We further enable BiGR to perform text-to-image generation, showcasing its potential for broader applications.

---

*Project lead

†Corresponding authors

# 1 INTRODUCTION

Image generation is experiencing a revolutionary growth driven by the advancements in diffusion models (Ho et al., 2020; Rombach et al., 2022; Peebles & Xie, 2023; Ma et al., 2024) and autoregressive models (Esser et al., 2021; Sun et al., 2024; Tian et al., 2024). While these models have demonstrated impressive performance, their representation capabilities are under-studied. As revealed by Balestriero & LeCun (2024), reconstruction-based learning often produces visually compelling results but fails to provide strong latent representations for perception. It has been a long-desired goal of the research community to design a good image generator which can also serve as a strong feature extractor.

Centered around this goal, previous studies (Chen et al., 2020a; Li et al., 2023a) on representation capabilities of generative models have primarily focused on unconditional generation. Despite conditional generation (Peebles & Xie, 2023; Sun et al., 2024; Li et al., 2024) has emerged as a recent research trend and garnered much attention, investigations of the representation capabilities of conditional generative models remain limited. In conditional image generation, conditions are added to guide the generation process. However, this guidance is absent in downstream discriminative tasks. This weakens the relationship between features and categories, and thereby diminishes the representation capabilities of the features. We validate this limitation using the latest class-conditional image generation model (Sun et al., 2024) (see Fig. 1 (right)), and stress the necessity of improving the representation capabilities of conditional generative models.

We introduce **BiGR**, a novel conditional image generation model that utilizes compact **Bi**nary latent codes for **G**enerative tasks with improved **R**epresentation capabilities. BiGR is trained exclusively through a generative process by reconstructing tokens without relying on any discriminative losses. We compress an image into a sequence of binary codes using lookup-free quantization (Yu et al., 2024; Wang et al., 2023) and utilize our model to predict these binary codes. We emphasize that BiGR is the first conditional image generation model that unifies generative and discriminative tasks, achieving improved performance across both. Below, we describe our model design, generative and discriminative use, and zero-shot generalized applications.

Our framework, built upon the language model architecture, has three major components, namely **(1)** a binary tokenizer that converts a pixel-level image into a sequence of binary latent codes, **(2)** a transformer equipped with full bidirectional attention, and **(3)** a binary transcoder that transforms continuous features into Bernoulli-distributed binary codes. We train BiGR using the masked modeling approach (Bao et al., 2022; Chang et al., 2022; Li et al., 2023a). This modification, deviating from the typical autoregressive approach, expands token interaction without altering the structure of Llama. Paired with a tailored inference process and inherent visual representations, BiGR can perform both generative and discriminative tasks.

For *generative* purpose, we design a sampling method that iteratively unmask tokens in a sequence, with the order determined by the binary entropy magnitude from the predicted Bernoulli distribution probabilities. This approach requires only a small number of sampling iterations which significantly accelerates the inference process. As a result, we achieve high efficiency in image generation compared with diffusion models, which require multiple steps to remove noise. Through extensive experiments, we show that BiGR performs on par with, or even surpasses, existing baselines in quantitative metrics.

For *discriminative* purpose, we perform average pooling on the intermediate features in BiGR. By this straightforward operation, BiGR exhibits significantly stronger representation capabilities than comparable models, which has been empirically validated through linear probe evaluation. Due to the compactness of binary codes and the global information from masked modeling, the feature representations produced by BiGR can more effectively linearly separate visual categories in downstream discriminative tasks.

Moreover, we explore the zero-shot generalization capabilities of BiGR within the generation domain. Unlike autoregressive models that must predict tokens in raster order, the masked modeling mechanism offers a huge flexibility during inference, allowing for the design of task-specific strategies. As a result, BiGR can perform various vision tasks in a zero-shot manner, without requiring any structural changes or parameter fine-tuning. In this paper, we showcase applications of our model in image inpainting, outpainting, editing, interpolation, and enrichment. We further extend BiGR to perform text-to-image generation, highlighting its potential for broader applications. The

generated results are provided in the Appendix. We believe that further applications of BiGR can be unlocked through community efforts.

To summarize, our BiGR possesses the following prominent advantages: **(i) Uniformity** - BiGR is the first conditional image generation model that unifies generative and discriminative tasks within the same model. By modeling compact binary latent codes, BiGR delivers strong performance in both tasks compared to existing models. **(ii) Efficiency** - BiGR generates images at a low time cost, attributed to the small number of sampling steps required in the iterative unmasking process, while still maintaining high generation quality. **(iii) Flexibility** - BiGR can be flexibly employed for various vision applications, such as inpainting, outpainting, editing, interpolation, and enrichment in a zero-shot manner, without the need for task-specific structural changes or parameter fine-tuning. **(iv) Scalability** - BiGR demonstrates scalability in both generative and discriminative tasks, as evidenced by comprehensive evaluations of both generation quality and linear-probe performance.

## 2 RELATED WORK

**Binary latent code modeling**  Binary latent code, also known as hashing (Wang et al., 2017), has demonstrated great effectiveness in visual representations due to its compactness and discreteness (Cakir et al., 2019; Jiang & Li, 2018; Shen et al., 2015; Wei et al., 2021; Wu et al., 2019). In the realm of visual generation, the study of binary tokenizers has recently attracted notable attention, referred as look-up free quantization in Yu et al. (2024) and as binary autoencoder in Wang et al. (2023). Binary tokenizers can enhance the codebook utilization for vector-quantization methods (Esser et al., 2021; Van Den Oord et al., 2017), facilitating image and video generation. Wang et al. (2023) introduces a Bernoulli diffusion process that operates on Bernoulli-distributed variables to generate binary latents. Our work studies this type of tokenizers and we propose a novel generative framework for uniform conditional generation and visual representation.

**Generative representation learning**  Representation learning has long been an important topic, with self-supervised methods (He et al., 2020; Chen et al., 2020b; Caron et al., 2020; Grill et al., 2020; Caron et al., 2021; Zhou et al., 2022) dominating the field in the past few years. Some works learn visual representations through generative modeling. For example, iGPT (Chen et al., 2020a) predicts pixels in a manner similar to GPTs (Brown et al., 2020), while MAE (He et al., 2022) and MAGE (Li et al., 2023a) reconstruct masked image regions. ViT-VQGAN (Yu et al., 2022a) studies the representation capabilities of unsupervised generative models. However, these methods involve specialized designs for discriminative tasks and are not directly suited for conditional image generation. Our work broadens this scope by proposing a conditional image generation framework that consistently delivers both high-quality generation and strong representation capabilities.

**Conditional image generation**  Conditional image generation has gained significant attention recently. Existing works on this topic can be broadly grouped into two categories: *diffusion* models (Ho et al., 2020; Song et al., 2021; Rombach et al., 2022; Peebles & Xie, 2023; Ma et al., 2024; Chen et al., 2024b), which gradually denoise a random Gaussian noise, and *autoregressive* models (Esser et al., 2021; Yu et al., 2022b;a; Sun et al., 2024; Tian et al., 2024), which predict the next tokens similarly to language models. The models based on masked prediction (Chang et al., 2022; Li et al., 2023a; Chang et al., 2023) can be classified as autoregressive models, as discussed in (Li et al., 2024). In this paper, for clarity, we use "autoregressive" to specifically refer to models that use causal attention and next-token prediction, and "mask" to refer to models using masked modeling. Although conditional generative models can produce visually compelling images, their representation capabilities have rarely been studied. Our work aims to bridge this gap.

## 3 METHOD

Our framework is based on a masked language model that operates directly on binary latent codes derived from images. We train the model by masking a portion of the input tokens and learning to unmask them using predicted output tokens. The prediction is achieved through a Bernoulli diffusion process (Wang et al., 2023), which is well-suited for generating binary latent codes. In sampling, we determine the order of tokens to be unmasked based on the magnitude of entropy computed from the predicted Bernoulli distribution probabilities. To obtain latent representations, we perform average pooling on the intermediate features of our model. We present the overview of BiGR in Fig. 2. We describe the details of each of its components below.

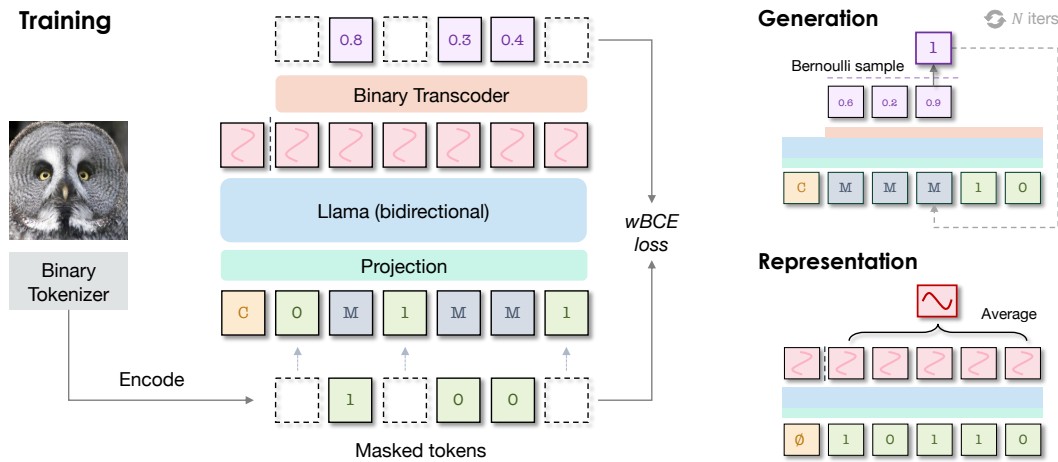

Figure 2: **Overview of BiGR.** For simplicity, we display only 1 bit for each token, although each token actually consists of K bits in length. **Left**: We outline the training of BiGR. Starting with binary codes from binary tokenizers, we append a condition token and mask partial tokens. These tokens are projected into continuous embeddings and processed by the Llama backbone. The outputs undergo a Bernoulli denoising process in the binary transcoder to generate probabilities, penalized by the weighted binary cross-entropy loss (wBCE) at masked positions. **Right**: We illustrate the generation process (detailed in Sec. 3.3) and the representation acquisition via average pooling.

## 3.1 PRELIMINARY

We first review the binary tokenizer and the Bernoulli diffusion process that underpin our model.

**Binary tokenizer**  An image tokenizer $\mathcal{T}$ can encode an image $x \in \mathbb{R}^{3 \times H \times W}$ into a sequence of latent codes $\{\zeta^1, \zeta^2, \cdots, \zeta^n\} = \mathcal{T}(x)$, where each $\zeta^i$ represents the code at a specific spatial position. Binary tokenizers (Yu et al., 2024; Wang et al., 2023), also known as lookup-free quantization, transform the code into binary format by

$$z^i = \text{sign}(\zeta^i) = \mathbb{1}\{\zeta^i > 0\}, \tag{1}$$

and a corresponding token index $r^i$ can be computed by

$$r^i = \sum_{k=1}^{K} 2^{k-1} \cdot z^i[k], \tag{2}$$

where $z^i[k]$ denotes the $k$-th bit of the binary code $z^i$, and $K$ is the number of binary bits (*i.e.*, code dimension), resulting in a total of $2^K$ token indices. Using Eq. (2), Yu et al. (2024) indexes image tokens with the binary code $z^i$ and build a vocabulary of size $2^K$ for generative purposes. In contrast, our approach focuses on directly modeling the sequence of binary codes $\{z^i\}_{i=1}^n$.

**Bernoulli diffusion**  We generate binary codes through a Bernoulli diffusion process (Wang et al., 2023), which effectively models Bernoulli-distributed variables. Specifically, Bernoulli diffusion process adds Bernoulli noise from the starting point $z \sim q(z^0)$:

$$q(z^t|z^{t-1}) = \mathcal{B}\left(z^t; z^{t-1}(1-\beta^t) + 0.5\beta^t\right) \quad t = 1, 2, \cdots, T. \tag{3}$$

Here, $\mathcal{B}$ denotes a Bernoulli distribution, and the timestep $t$ out of the total $T$ is denoted as a superscript. We model the denoising process by $p(z^{t-1}|z^t)$, which predicts the Bernoulli distribution probabilities for the binary code at the previous timestep. By iterating the denoising process, starting with a coin toss $\mathcal{B}(0.5)$, we can finally generate binary codes that follow Bernoulli distributions. In our model, the **binary transcoder** is the component that integrates the Bernoulli diffusion process, responsible for denoising Bernoulli noise and predicting binary codes.

## 3.2 MASKED MODELING ON BINARY LATENT CODES

**Backbone**  We build our method upon the transformer-based language model Llama (Dubey et al., 2024; Touvron et al., 2023b;a). Unlike language, an image is not naturally modeled as a causal sequence of tokens, but instead, each token should have access to all others to better capture global

visual information. Therefore, we replace the causal attention commonly used in language models with bidirectional attention, and let the model predict masked tokens instead of next tokens.

**Input projection** In the input space, instead of looking up an embedding vector with a token index, we use a simple linear layer that projects the binary code onto the embedding space. This technique has recently been explored for continuous-valued tokenizers in Tschannen et al. (2023), and we find that it also works well for binary-valued tokenizers. We maintain standard conditional embeddings and mask embeddings, where the conditional embedding is appended at the start of the sequence, and the mask embedding replaces inputs at masked positions.

**Mask-token prediction** During training, we simply mask a portion of image tokens with a learnable mask token [M]. The fraction of masked tokens follows a cosine schedule, as used in Li et al. (2023a). We compute losses only for the masked positions, where the model predicts the values of the masked tokens. Formally, let $f_\theta$ represent the language model, and $\{z_{m_i}^i\}_{i=1}^n$ denote the sequence of binary codes that are partially masked. Here, $M = \{m_i\}_{i=1}^n$ indicates whether the $i$-th position is masked ($m_i = 1$) or left unmasked ($m_i = 0$). We obtain outputs at the masked positions from the language model $\{h^i\}_{m_i=1} = f_\theta(\{z_{m_i}^i\}_{i=1}^n)$, which are distributed in a continuous space.

**Binary transcoder** We transform the model outputs $h$ into binary codes[1] $z$ through a Bernoulli diffusion process (Wang et al., 2023). In particular, we learn a denoising network $g_\phi$ with a Sigmoid function $S$ to model

$$p_\phi(z^{t-1}|z^t) = \mathcal{B}\left(z^{t-1}; S(g_\phi(z^t, t, h))\right), \tag{4}$$

which predicts the probabilities of the Bernoulli distribution conditioned on the intermediate feature $h$. Consequently, binary codes can be generated by sampling from these probabilities. Following Wang et al. (2023); Ho et al. (2020), the training target is the binary residual, *i.e.*, $z^t \oplus z^0$ where $\oplus$ represents the element-wise XOR operation. The training objective is simply an element-wise weighted binary cross-entropy (wBCE) loss, expressed as

$$y_k = (z^t \oplus z^0)[k] \in \{0, 1\} \quad p_k = S(g_\phi(z^t, t, h))[k] \in [0, 1], \tag{5}$$

$$\mathcal{L} = -\frac{1}{K} \sum_{k=1}^K w_k \left(y_k \log p_k + (1 - y_k) \log(1 - p_k)\right), \tag{6}$$

$$\text{where} \quad w_k = (1 - y_k) \cdot \sum_{k=1}^K y_k/K + y_k \cdot (1 - \sum_{k=1}^K y_k/K) + 1/K. \tag{7}$$

Here, $y_k$ represents the target, and $p_k$ is the predicted probability for the $k$-th bit in the binary code. The element-wise loss weight $w_k$ is applied to mitigate the imbalance between 0s and 1s, calculated based on their respective ratios in a $K$-dimensional code. The constant term $1/K$ is added to prevent nearly-zero weights that could impede training. In training, we jointly optimize the language model $f_\theta$ and the denoising network $g_\phi$ using the loss defined in Eq. (6).

**Visual representation** Once trained, our model inherently possesses strong visual representations. Given an image, we input it into the model without any masks, along with an unconditional token appended. We then perform average pooling on the continuous-valued features $h$ to derive the global representation of the given image. We observe that the most discriminative representation originates not from the final layer but from the middle layers within the transformer blocks, in line with the findings in Yu et al. (2022a); Chen et al. (2020a). As a result, we use the intermediate features as the final image representation.

### 3.3 Entropy-ordered generative sampling

For image generation, we design a sampling strategy for our model, enabling it to iteratively predict tokens from a fully masked sequence. Unlike in training, where mask positions are randomly chosen at each step, during sampling, the order in which tokens are unmasked follows a predefined criterion.

We arrange the masked tokens according to the binary entropy magnitude calculated from the predicted probabilities. The binary entropy is defined as:

$$\mathcal{H} = -\frac{1}{K} \sum_{k=1}^K p_k \log_2 p_k + (1 - p_k) \log_2(1 - p_k), \tag{8}$$

---

[1]Since the operation is position-wise, we omit the superscript of positions for simplicity.

which ranges from 0 to 1. Here, a low value indicates high prediction confidence (*i.e.*, when $p_k$ is closer to either 1 or 0). Therefore, a confidence score can be derived from $1 - \mathcal{H}$, illustrating the model's confidence in this prediction. Following Li et al. (2023a), we add a noise sampled from a random Gumbel distribution multiplied by the temperature $\tau$ to the confidence score.

At each sampling iteration, we select and unmask a proportion of masked positions with the highest confidence scores. To unmask each token, we obtain its binary codes by performing Bernoulli sampling from the distribution $\mathcal{B}(p_k)$. The unmasking ratio follows a cosine schedule as used in Chang et al. (2022); Li et al. (2023a). This process operates over $N$ sampling iterations. When the mask ratio drops to zero, the sampling progresses to the last iteration where all tokens are unmasked, marking the completion of the generation process.

# 4 EXPERIMENT

## 4.1 IMPLEMENTATION DETAILS

**Model configuration** We use the binary autoencoder (B-AE) introduced by Wang et al. (2023) as our binary tokenizer. The downsampling rate of the autoencoder is 16, projecting a 256×256 image into a 16×16 token sequence. We train four variants of the binary autoencoders designed with four different binary code dimensions, namely 16, 20, 24, and 32. With the four binary tokenizers, we train our BiGR of three different sizes based on Llama (Touvron et al., 2023a), namely L (316M), XL (743M), and XXL (1.38B). For the binary transcoder, we follow Li et al. (2024) to employ an MLP $g_\phi$ with an adaptive LayerNorm, with sizes of 20M, 56M, and 104M respectively. For clarity, we denote the S-sized variant with a B-dim tokenizer as BiGR-S-dB, *e.g.*, BiGR-L-d16.

**Training details** We train all models for 400 epochs, with L-sized models using a batch size of 1024 and the others using a batch size of 512. Our L/XL-sized models are trained on 8 A800 GPUs, while XXL-sized models are trained on 32 A800 GPUs. We maintain consistent training settings across all compared models based on the model size.

**Sampling** Our model inherently supports classifier-free guidance (CFG) (Ho & Salimans, 2022) through the Bernoulli diffusion process. Within our sampling process, four hyperparameters are involved: CFG scale, Gumbel temperature ($\tau$), the number of sampling iterations ($N$), and the number of Bernoulli denoising steps ($T$). We identify the optimal hyperparameter setting for all models. We set CFG to 2.5 for all quantitative evaluations, which has shown to be effective across all our models. We set $T$ to 100 as default for all models. See more details in Appendix A.

## 4.2 UNIFORM PERFORMANCE

**Evaluation** We evaluate the uniformity of BiGR by concurrently comparing generative and discriminative performance. We evaluate **generation quality** on ImageNet-1K 256×256 by reporting Frechet Inception Distance (FID) as the main metric, along with Inception Score (IS), sFID, Precision (Pre.), and Recall (Rec.) as auxiliary metrics. All metrics are obtained using 50K generated samples. We assess **representation capabilities** through linear-probe evaluation, reporting the top-1 and top-5 accuracies, abbreviated as ACC1 and ACC5, on ImageNet-1k 256×256 validation split. We follow standard practice (He et al., 2022) by using a parameter-free BatchNorm (Ioffe, 2015) layer and a linear classifier layer to classify the model features. We use the intermediate features from the 10-th layer for L-sized models, the 15-th layer for XL-sized models, and the 32-nd layer for XXL-sized models, as experiments on d16 models demonstrate these configurations yield the best performance. Additionally, we compare the inference speed, specifically the time taken to generate each image using one 4090 GPU with a batch size of 64.

**Comparison** Starting from the latest autoregressive generation baseline LlamaGen (Sun et al., 2024), we comprehensively analyze two major components in this paper, namely (1) training objectives, specifically categorical loss (cat.) and binary loss (bin.), and (2) modeling types, including masking and autoregressive (AR) approaches. In total, we compare five models in Tab. 1, training four models—S0, S1, S2, and S3—with different configurations, excluding LlamaGen. For Llama-Gen, we use the generative performance results under a *fair* setting of 256×256 image generation, as reported in their paper. We conduct our own evaluation of linear-probe performance using their pretrained model. The inference time of all models is tested on the same machines by us.

Table 1: **Uniformity comparison.** We compare the generative and discriminative performance of our model against LlamaGen (Sun et al., 2024) and three other settings, varying by tokenizers, training objectives, and modeling types. We use KV cache to accelerate all AR models.

| Model | Tokenizer | Objective | Type | Time↓ | Generative | | | | | Discriminative | |
|---|---|---|---|---|---|---|---|---|---|---|---|
| | | | | | FID↓ | IS↑ | sFID↓ | Pre.↑ | Rec.↑ | ACC1 | ACC5 |
| LlamaGen | VQGAN | Cat. | AR | 0.13 | 3.81 | 248.28 | 8.49 | 0.83 | 0.52 | 40.5 | 64.4 |
| S0 | B-AE | Cat. | AR | 0.15 | 3.21 | 239.17 | **5.38** | 0.83 | **0.54** | 23.8 | 44.2 |
| S1 | B-AE | Cat. | Mask | **0.10** | 3.85 | 261.81 | 6.10 | 0.85 | 0.47 | 61.1 | 83.2 |
| S2 | B-AE | Bin. | AR | 1.04 | 7.50 | 164.31 | 6.56 | 0.85 | 0.41 | 45.2 | 69.3 |
| S3 (Ours) | B-AE | Bin. | Mask | 0.69 | **3.17** | **262.14** | 5.59 | **0.86** | 0.50 | **64.3** | **85.4** |

Table 2: **Binary transcoder comparison.**

| Binary objective | Generative | | | | | Discriminative | |
|---|---|---|---|---|---|---|---|
| | FID↓ | IS↑ | sFID↓ | Pre.↑ | Rec.↑ | ACC1 | ACC5 |
| *w/o Bernoulli denoising* | | | | | | | |
| Direct BCE | 5.84 | 212.34 | 9.89 | 0.78 | **0.52** | 63.3 | 84.8 |
| *w/ Bernoulli denoising* | | | | | | | |
| Predict $z^0$ | 4.39 | **274.26** | 9.07 | **0.87** | 0.44 | 62.0 | 83.9 |
| Predict $z^t \oplus z^0$ (Ours) | **3.17** | 262.14 | **5.59** | 0.86 | 0.50 | **64.3** | **85.4** |

Table 3: **Sampling order comparison.** We include the autoregressive variant for reference.

| Type | Order | Time↓ | FID↓ | IS↑ | sFID↓ | Pre.↑ | Rec.↑ |
|---|---|---|---|---|---|---|---|
| AR | Raster | 1.04 | 7.50 | 164.31 | 6.56 | 0.85 | 0.41 |
| Mask | Raster | 8.81 | 4.51 | 191.10 | 6.49 | 0.80 | 0.54 |
| Mask | Rand. | **0.69** | 7.12 | 174.11 | 11.85 | 0.76 | **0.55** |
| Mask | Ours | **0.69** | **3.17** | **262.14** | **5.59** | **0.86** | 0.50 |

**Observation**   As shown in Tab. 1, our model significantly outperforms other methods across all main evaluation metrics. In addition, we have the following observations. **(1)** By comparing LlamaGen and S0, using binary autoencoder provides better generative performance and worse discriminative performance compared to VQGAN. **(2)** For generation, AR modeling is better suited for categorical loss, while masked modeling is more appropriate for binary loss. **(3)** For discrimination, masked modeling drastically outperforms AR modeling for both losses, with binary loss further enhancing performance. **(4)** Masked modeling achieves significantly faster inference speed compared to AR modeling due to its fewer sampling iterations, with the binary objective taking more time resulting from the diffusion process. This can be further accelerated by reducing sampling iterations and diffusion timesteps, as discussed in Fig. 3. To conclude, BiGR, which employs masked modeling on binary latent codes, achieves the best *uniform* performance on both generative and discriminative tasks, accompanied by an efficient inference runtime.

## 4.3   MODEL ANALYSIS

We analyze each component of our proposed method below. All experiments are conducted on BiGR-L-d16 unless otherwise specified.

**Binary transcoder**   We apply Bernoulli denoising process (Wang et al., 2023) as our binary transcoder to generate probabilities of Bernoulli distributions, from which the binary codes are sampled. We experiment with two variants, namely (1) predicting the initial clean latent $z_0$, and (2) predicting the element-wise exclusive OR (XOR) value between the latent at the $t$-th timestep $z^t$ and $z^0$. We find empirically the latter performs better, and thus, we adopt this setting for all of our models. Alternatively, a naïve approach involves using a direct binary cross-entropy (BCE) loss to train the model, replacing the Bernoulli denoising process. We compare these three variants in Tab. 2. Our method outperforms the other two variants across all main metrics. We observe that using direct BCE generates very smooth images which harms the generative performance. XOR prediction yields better generative and discriminative performance compared to $z^0$ prediction.

**Sampling strategy**   In this paper, we propose a simple entropy-ordered sampling strategy tailored for the masked training paradigm. We compare our method with two alternative sampling orders, namely (1) a raster-scan order similar to the autoregressive approach, and (2) a random order. Like our strategy, both compared methods are applied to the same trained model. The comparison results of the generative evaluation are reported in Tab. 3. The results indicate that the proposed sampling strategy is the best fit for our model's generative purposes.

**Inference hyperparameters**   We evaluate the impact of two hyperparameters specific to our model on its performance. **(1)** We first present the FID results and sample time for different numbers of sampling iterations $N$ on the left side of Fig. 3. We observe that larger models generally achieve lower FID values, although they also increase sample time. In addition, more sampling iterations do not guarantee better performance, as different-sized models have varying optimal sampling iterations. For example, the L-sized model achieves its best performance with 20 iterations, rather than with larger numbers. **(2)** On the right side of Fig. 3, we present the results for different numbers of diffusion timesteps $T$. The results indicate that diffusion timesteps have a marginal impact on

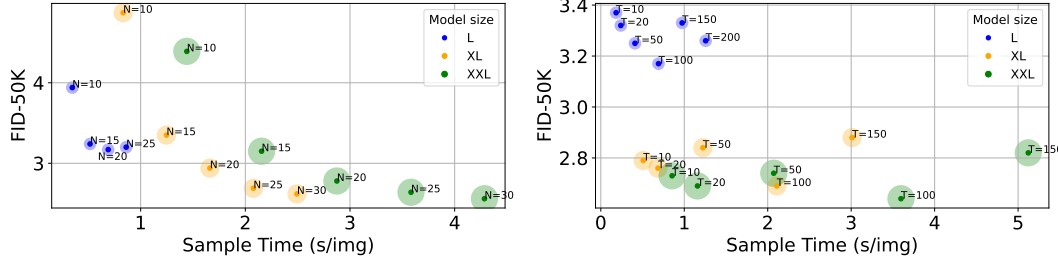

Figure 3: **Relationships between FID-50K and sample time across varying inference hyper-parameters.** We compare different numbers of sampling iterations $N$ (left) and varying diffusion timesteps $T$ (right) for three model sizes. All other hyperparameters are kept at their default settings.

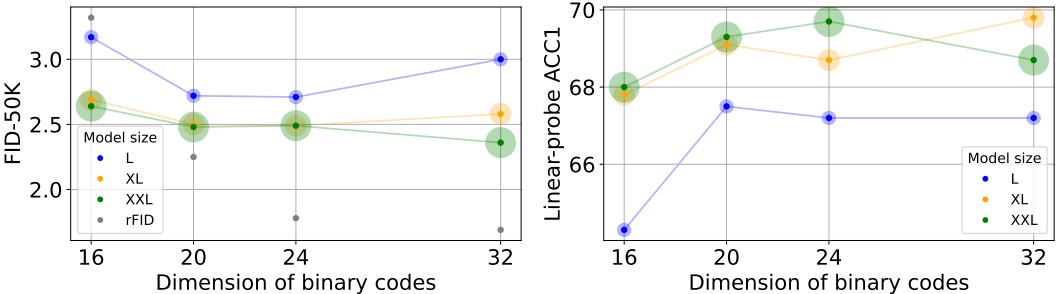

Figure 4: **Evaluation of generative and discriminative performance across different model sizes.** We report results for all tested tokenizers across four different dimensions of binary codes. We include the reconstruction FID (rFID) for each binary tokenizer for reference (grey points).

generative performance, suggesting that our model can achieve comparable generation quality with fewer diffusion timesteps. This can significantly accelerate generation speed, especially for larger models. For example, with 10 diffusion timesteps, the XXL-sized model can achieve an FID of 2.73 at a speed of 0.85s per image.

**Model size and code dimension** We validate that our model is scalable by testing the performance of different-sized models using tokenizers with various code dimensions. Note that the dimension of the binary codes only alters the number of parameters in the input and output linear projections, resulting in minimal effects on the overall model size. The evaluation results of both generative and discriminative performance are shown in Fig. 4. Our model generally performs better with larger sizes across all code dimensions, as indicated by both generative and discriminative metrics.

Besides, we have the following observations from Fig. 4. **(1)** When the model size is small, it becomes challenging to model large-dimensional codes, such as a dimension of 32 for the L-sized model, especially for generative purpose. **(2)** In contrast, as the model size increases, the improvement for smaller-dimensional codes is relatively modest, indicating that these codes are easier to model and can be effectively handled by smaller-sized models. **(3)** An exception arises in the linear-probe evaluation of models with 32-dimensional codes, where our XL-sized model outperforms the XXL-sized model. We hypothesize that this may be due to the optimal transformer layer for feature representation identified in the 16-dimensional model, which might not be the best choice for 32-dimensional models of the XXL size.

**Unconditional training** Our model is a class-conditional generative model. Intuitively, conditional generative training adds condition guidance that is absent in downstream discriminative tasks, which can diminish the representation capabilities of the model. We validate this conjecture by comparing the linear-probe performance of our model with that of its unconditional counterpart. We train the unconditional model by replacing the class conditional tokens with a single unconditional token, and keep the inference process unchanged. We evaluate BiGR-L-d20 alongside its unconditional counterpart, and report the results in Tab. 4. The unconditional counterpart demonstrates better representation capabilities than our conditional model, indicating that discriminative tasks are more challenging for conditional generative models.

**Resolution of 512×512** Using a binary autoencoder that projects a 512×512 image into 32×32 binary latent codes, we enable our model to generate 512×512 images by increasing the input se-

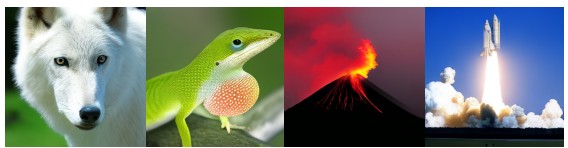

Figure 5: **Generated 512×512 samples.**

Table 4: **Linear-probe evaluation** of conditional and unconditional counterparts.

| Training | ACC1 | ACC5 |
|---|---|---|
| Cond. | 67.5 | 87.5 |
| Uncond. | **68.3** | **88.4** |

Table 5: **Generative performance comparison** on 256×256 ImageNet-1K benchmark.

| Type | Model | #Params. | FID↓ | IS↑ |
|---|---|---|---|---|
| Diff. | DiT-L/2 (Peebles & Xie, 2023) | 458M | 5.02 | 167.2 |
| | DiT-XL/2 | 675M | 2.27 | 278.2 |
| | SiT-XL/2 (ODE) (Ma et al., 2024) | 675M | 2.15 | 258.1 |
| | SiT-XL/2 (SDE) | 675M | 2.06 | 277.5 |
| Mask | MaskGIT (Chang et al., 2022) | 227M | 6.18 | 182.1 |
| AR | VQGAN (Esser et al., 2021) | 227M | 18.65 | 80.4 |
| | VQGAN | 1.4B | 15.78 | 74.3 |
| | ViT-VQGAN (Yu et al., 2022a) | 1.7B | 4.17 | 175.1 |
| | RQTran. (Lee et al., 2022) | 3.8B | 7.55 | 134.0 |
| VAR | VAR-d16 (Tian et al., 2024) | 310M | 3.30 | 274.4 |
| | VAR-d20 | 600M | 2.57 | 302.6 |
| | VAR-d24 | 1.0B | 2.09 | 312.9 |
| | VAR-d30 | 2.0B | 1.92 | 323.1 |
| MAR | MAR-B (Li et al., 2024) | 208M | 2.31 | 281.7 |
| | MAR-L | 479M | 1.78 | 296.0 |
| | MAR-H | 943M | 1.55 | 303.7 |
| AR | LlamaGen-B (Sun et al., 2024) | 111M | 5.46 | 193.6 |
| | LlamaGen-L | 343M | 3.81 | 248.3 |
| | LlamaGen-XL | 775M | 3.39 | 227.1 |
| | LlamaGen-XXL | 1.4B | 3.09 | 253.6 |
| | LlamaGen-3B | 3.1B | 3.05 | 222.3 |
| Ours | BiGR-L-d24 | 336M | 2.71 | 275.7 |
| | BiGR-XL-d24 | 799M | 2.49 | 278.8 |
| | BiGR-XXL-d32 | 1.5B | 2.36 | 277.2 |

Table 6: **Linear-probe top-1 accuracy** on ImageNet-1K. [†]: our evaluation results.

| Type | Method | #Tokens | Params | ACC1↑ |
|---|---|---|---|---|
| Con. | MoCo (He et al., 2020) | - | 375M | 68.6 |
| | SimCLR (Chen et al., 2020b) | - | 375M | 76.5 |
| | SwAV (Caron et al., 2020) | - | 93M | 75.3 |
| | DINO (Caron et al., 2021) | - | 85M | 75.3 |
| | BYOL (Grill et al., 2020) | - | 375M | 78.6 |
| | CAE (Chen et al., 2024c) | - | 304M | 78.1 |
| | CMAE (Huang et al., 2023) | - | 86M | 73.9 |
| MIM | iBOT (Zhou et al., 2022) | - | 304M | 81.0 |
| | BEiT (Bao et al., 2022) | 16×16 | 307M | 73.5 |
| | MAE (He et al., 2022) | 14×14 | 304M | 75.8 |
| | MAGE (Li et al., 2023a) | 16×16 | 328M | 78.9 |
| Gen. | BigBiGAN (Brock, 2018) | - | 344M | 61.3 |
| | iGPT-L (Chen et al., 2020a) | 32×32 | 1.4B | 60.3 |
| | iGPT-L | 48×48 | 1.4B | 65.2 |
| | ViT-VQGAN-B (Yu et al., 2022a) | 32×32 | 650M | 65.1 |
| | ViT-VQGAN-L | 32×32 | 1.7B | 73.2 |
| | RCG (Li et al., 2023b) | 16×16 | 304M | 77.6 |
| | l-DAE (Chen et al., 2024d) | - | 304M | 75.0 |
| Cond. gen. | LlamaGen-L[†] (Sun et al., 2024) | 16×16 | 343M | 40.5 |
| | MAR-B[†] (Li et al., 2024) | 16×16 | 208M | 57.9 |
| | MAR-L[†] | 16×16 | 479M | 59.1 |
| | MAR-H[†] | 16×16 | 943M | 60.0 |
| | BiGR-L-d20 (Ours) | 16×16 | 336M | 67.5 |
| | BiGR-XL-d32 (Ours) | 16×16 | 799M | 69.8 |

quence length to 1024. We train such a binary autoencoder with a code dimension of 32 and train our model to accommodate this sequence length. We showcase the generated samples in Fig. 5, with additional samples available in Appendix F.

## 4.4 SYSTEM-LEVEL COMPARISONS

We re-emphasize that the goal of this work is to propose a uniform conditional generative model that can produce high-quality generations while maintaining strong representation capabilities. Therefore, surpassing state-of-the-art models across all metrics is not within the scope of this research. We provide a more comprehensive comparison in Appendix E.

**Conditional image generation** We present a comparison of the generative performance of our model with leading generative systems in Tab. 5. The recent MAR (Li et al., 2024) achieves the lowest FID, and SiT (Ma et al., 2024) sets the state-of-the-art among diffusion-based models. Our model maintains top-tier generative quality among the first echelon of approaches. Besides, BiGR significantly outperforms LlamaGen.

**Visual representation** We compare the linear-probe results of our model and the previous methods specifically designed for discriminative tasks. The results are shown in Tab. 6. We categorize the compared models into several types: contrastive (Con.), masked image modeling (MIM), generative (Gen.), and conditional generative (Cond. gen.). This classification is not entirely precise, as some models may use multiple losses for training, like MAGE (Li et al., 2023a). Our model is fairly compared to the conditional generative models, which solely rely on plain reconstruction loss without discriminative designs, such as specialized losses, augmentations, or additional data. For LlamaGen (Sun et al., 2024) that has the same model architecture as ours, we use the feature from the same layer for linear layer training. For MAR (Li et al., 2024), since their structure largely resembles MAE (He et al., 2022), we follow MAE's approach and train the linear layer on top of the encoder outputs. BiGR significantly outperforms the other conditional generative models.

## 4.5 ZERO-SHOT GENERALIZED APPLICATIONS

The nature of the masked modeling mechanism allows the use of BiGR in a wide range of applications in a zero-shot manner, without the need for task-specific structural changes or parameter fine-tuning. We present the results of BiGR applied across various tasks in Fig. 6.

| Inpainting | Outpainting | Editing | Interpolation | Enrichment |

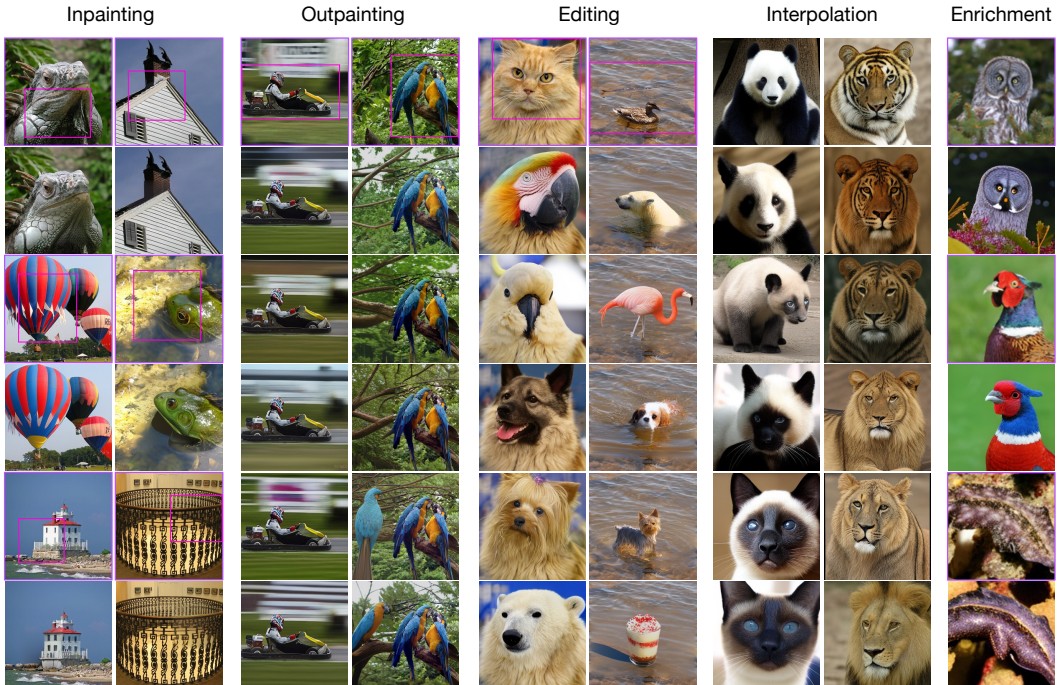

Figure 6: **Zero-shot generalization.** We present samples of inpainting, outpainting, editing, interpolation, and enrichment. The original image is marked with a purple border, with a pink box highlighting the masked region. Images without the purple borders are generated by our model.

**Inpainting & Outpainting**   Given an image with a mask, we use the unmasked regions to initialize the model inputs, enabling it to generate the remaining masked tokens. This generation process is guided by an unconditional token, which ensures that no additional information is introduced, allowing the model to focus solely on the existing image information. This approach enables high-quality and diverse inpainting and outpainting.

**Class-conditional editing**   Unlike inpainting and outpainting, class-conditional editing is guided by a specific class condition, allowing the model to edit the masked region with a designated class object. Other operations remain consistent with inpainting and outpainting.

**Class interpolation**   We interpolate between two class conditions by calculating a weighted sum in the embedding space. We then use the resulting interpolated embedding to guide the generation process. This interpolation process demonstrates that our model can generalize visual characteristics across different classes rather than merely memorizing each class.

**Image enrichment**   Our model can also enrich visual details in a low-resolution image, a process we call image enrichment. Specifically, we first upsample a $128\times128$ image to a resolution of $256\times256$ and encode it into a sequence of $16\times16$ tokens. This approach leverages the model's generative capabilities to enrich images from low-resolution inputs.

## 5   CONCLUSION

We introduce BiGR as the first conditional generative model that unifies generative and discriminative tasks within the same framework. Through extensive experiments, we highlight its uniformity, efficiency, flexibility, and scalability. Our results demonstrate that BiGR achieves decent performance in both generation quality and linear separability. Additionally, we showcase its application in various zero-shot generalized tasks. We believe BiGR has the potential to be adapted for a broader range of applications in the future.

**Limitations**   (1) Our sampling strategy involves numerous hyperparameters to tune, resulting in a substantial search space; thus, the reported models may not represent the optimal settings. (2) The model's sequence length is fixed during training, making it inflexible to accommodate inputs of varying lengths. Consequently, generating higher-resolution images requires re-training the model.

**Acknowledgement**   This work is partially supported by the Hong Kong Research Grants Council - General Research Fund (Grant No.: 17211024).

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

## A    ADDITIONAL IMPLEMENTATION DETAILS

**Model configuration**    The configuration settings for the model architecture, training and inference of BiGR across different model sizes are provided in Tab. 7. Unless otherwise specified, our model uses this default setting in the main paper. For the inference hyperparameters, we observe the following patterns:

1. **CFG scale**: A larger CFG scale produces smoother images with clearer class features, while a smaller scale enhances fine-grained details.

2. **Gumbel temperature**: A higher Gumbel temperature increases generation diversity but reduces quality, whereas a lower temperature improves quality at the cost of diversity.

3. **Sampling iterations**: 20–30 iterations generally perform well. Using 10 iterations speeds up generation but slightly reduces quality, while more than 30 iterations has minimal impact but slows down generation.

4. **Diffusion timesteps**: 100 steps typically yield good results. Performance remains largely consistent across a broad range of 10–200 steps, with only marginal differences.

Table 7: **The default configuration settings of three models: BiGR-L, BiGR-XL, BiGR-XXL.**

| BiGR-L | | BiGR-XL | | BiGR-XXL | |
|---|---|---|---|---|---|
| Config | Value | Config | Value | Config | Value |
| Architecture | | Architecture | | Architecture | |
| Transformer layers | 24 | Transformer layers | 36 | Transformer layers | 48 |
| Transformer heads | 16 | Transformer heads | 20 | Transformer heads | 24 |
| Transformer dimensions | 1024 | Transformer dimensions | 1280 | Transformer dimensions | 1536 |
| MLP layers | 3 | MLP layers | 6 | MLP layers | 8 |
| MLP dimensions | 1024 | MLP dimensions | 1280 | MLP dimensions | 1536 |
| Training | | Training | | Training | |
| Batch size | 1024 | Batch size | 512 | Batch size | 512 |
| Epochs | 400 | Epochs | 400 | Epochs | 400 |
| Weight decay | 2e-2 | Weight decay | 2e-2 | Weight decay | 2e-2 |
| Learning rate | 1e-4 | Learning rate | 1e-4 | Learning rate | 1e-4 |
| Total diffusion timesteps | 256 | Total diffusion timesteps | 256 | Total diffusion timesteps | 256 |
| Inference | | Inference | | Inference | |
| CFG scale | 2.5 | CFG scale | 2.5 | CFG scale | 2.5 |
| Sampling iterations | 20 | Sampling iterations | 25 | Sampling iterations | 25 |
| Gumbel temperature | 0.17 | Gumbel temperature | 0.25 | Gumbel temperature | 0.30 |
| Diffusion timesteps | 100 | Diffusion timesteps | 100 | Diffusion timesteps | 100 |

**Binary transcoder**    After producing Bernoulli distribution probabilities through Bernoulli denoising, there are two ways to obtain binary codes: deterministic and non-deterministic. For deterministic method, values are set to 1 if the probability exceeds 0.5, and 0 otherwise. In contrast, for non-deterministic methods, we sample directly from the Bernoulli distribution to obtain 0 and 1 values. We empirically compare these two methods, as shown in Tab. 8 and find that the non-deterministic approach slightly outperforms its deterministic counterpart. As a result, we adopt the non-deterministic approach for all models presented in the main paper.

Table 8: **Comparison of deterministic and non-deterministic sampling.**

| Determ. | FID↓ | IS↑ | sFID↓ | Pre.↑ | Rec.↑ |
|---|---|---|---|---|---|
| ✓ | 3.19 | 239.79 | 6.25 | 0.84 | **0.52** |
| ✗ (Ours) | **3.17** | **262.14** | **5.59** | **0.86** | 0.50 |

**Sampling strategy**    In our sampling strategy, the implementation of Eq. (8) in the main paper may encounter a "nan" issue caused by the logarithmic operation. Since we only need to compare the relative magnitudes of different entries, we can instead use a value with the same trend to mimic the exact confidence. We use $2 \times |p_k - 0.5|$ as the final confidence value in our implementation.

**Adaptive LayerNorm**    We empirically find that adaptive LayerNorm (adaLN) has a marginal effect on performance. Following the approach in Tian et al. (2024), we implement a shared adaLN

that uses a single MLP to obtain the shift, scale, and gate values for all transformer layers, which adds only a minimal number of parameters.

**Linear probe**    Following the linear-probe evaluation protocol outlined in (He et al., 2022), we use the LARS optimizer with a momentum of 0.9. We train the linear head for 100 epochs, using a batch size of 256 along with 8 gradient accumulation steps. We use a warm-up period of 10 epochs and set the learning rate to 0.1. An extra BatchNorm layer is added before the linear classifier, without affine transformation. We refrain from using mixup, cutmix, drop path, or color jittering, and the weight decay is set to zero. We use the same linear-probe setting for all compared models in the main paper.

## B    TEXT-TO-IMAGE GENERATION

We validate the effectiveness of BiGR for text-to-image generation (T2I).

**Model adaptation**    We follow the practice of LlamaGen to adapt our class-conditional image generation model into a text-to-image generation model. Most parts of our model remain unchanged, except for replacing the class-condition token embedding with the text token embedding extracted by a text encoder. We use T5-XXL (Raffel et al., 2020)[2] as our text encoder. The maximum text token length is set to 120, with left-padding applied during both training and inference. The extracted text token embeddings are projected through an additional MLP layer and appended to the patch tokens. Conditioning for image generation relies solely on these appended text token embeddings, without using adaptive LayerNorm or cross-attention. Note that this minimal model adaptation is intended to validate the model's capability for text-to-image generation. More refined designs could be implemented to enhance generation performance, which we leave for future research.

**Training dataset**    We train our T2I model on a 20M subset of LAION-400M (Schuhmann et al., 2021) and JourneyDB (Sun et al., 2023), a large-scale dataset containing 4M high-quality images annotated with corresponding text captions. The training images are center-cropped and resized to resolutions of 256×256 and 512×512.

**Implementation details**    We conduct experiments using BiGR-XL-d24. For simplicity, we train our T2I model in a single stage, unlike the two-stage approach used in LlamaGen. The T2I model is initialized with the pretrained weights of our class-conditional generation model. We train the model on 32 A800 GPUs in three stages: (1) We initially train the model for 250K steps with a batch size of 1024 using JourneyDB at a resolution of 256×256; (2) We then finetune it for 450K steps with a batch size of 256 on both datasets at a resolution of 512×512; (3) Finally, we further finetune the model for 500K steps on JourneyDB at a resolution of 512×512.

**Generation samples**    We present our text-to-image generation samples with short prompts in Fig. 7 and with long prompts in Fig. 8, showing that our model performs well with both short and long prompts. The prompts we use are selected from LlamaGen (Sun et al., 2024) and various T2I works, such as PixArt (Chen et al., 2024a), Imagen (Saharia et al., 2022), and DALL-E 3 (Betker et al., 2023). These prompts are unseen in the training set, demonstrating that our T2I model generalizes well. Current results reveal the strong potential of BiGR in text-to-image generation. Note that our training data, model size, and training duration are relatively limited, and we only train a 512×512 T2I model, which has a relatively low image resolution. We believe that scaling up training data and model size, along with exploring advanced techniques such as incorporating higher resolutions, fine-tuning autoencoders, and using images with higher quality, can further enhance BiGR's T2I performance. We will explore this in future research.

## C    VISUALIZATION OF THE SAMPLING ORDER

We propose an entropy-ordered sampling process in this paper. To explore this sampling order further, we visualize the generated results at different iterations within the process. The visualization of the process is presented in Fig. 9. We observe that early iterations capture class-level characteristics, while subsequent iterations generate finer object-related details. In the final stages, visual quality steadily improves.

---

[2]https://huggingface.co/google/t5-v1_1-xxl

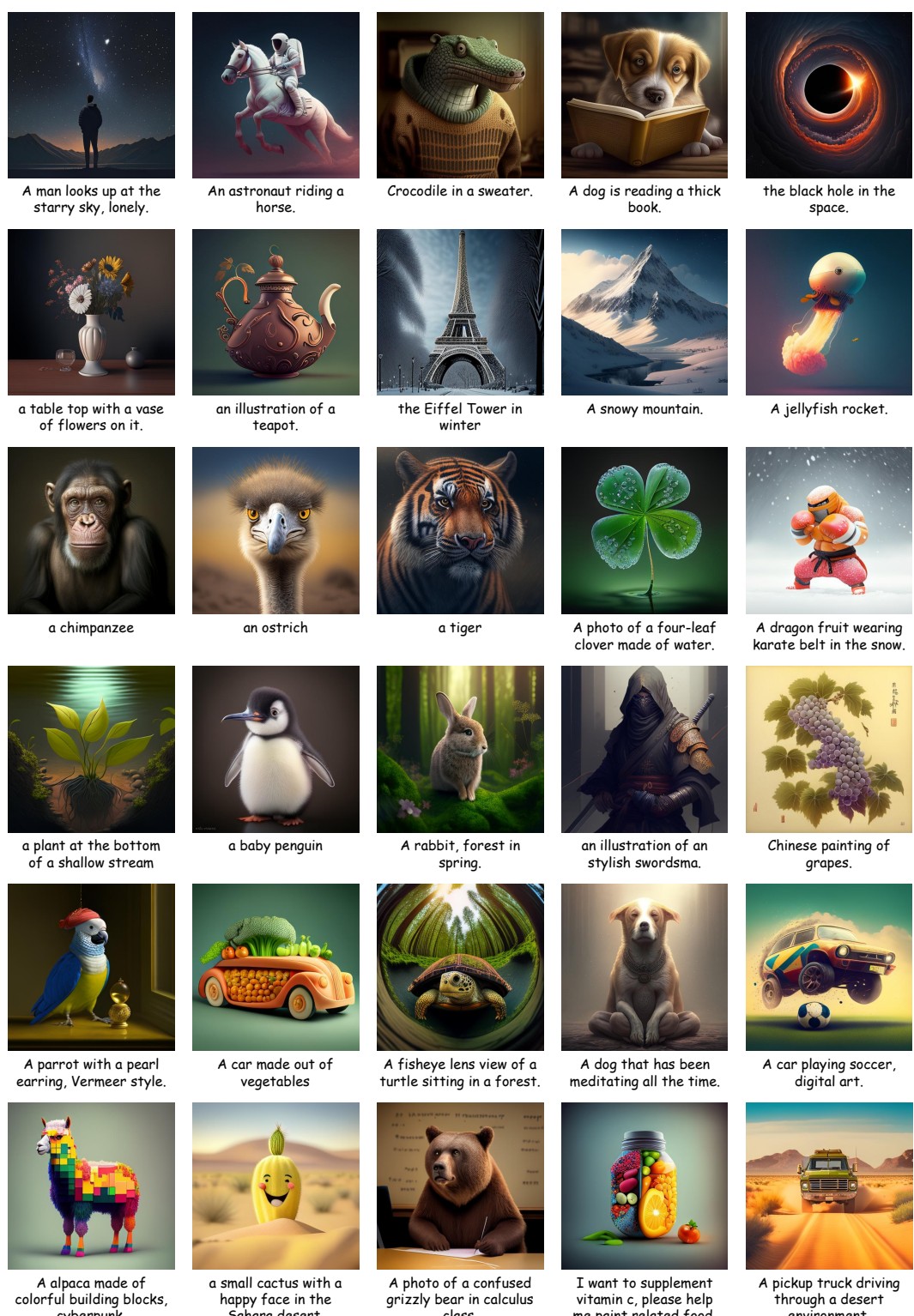

Figure 7: **Text-conditional 512×512 image generation samples with short prompts.** The results are generated by BiGR-XL-d24.

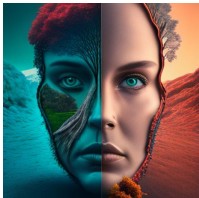

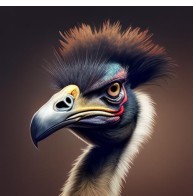

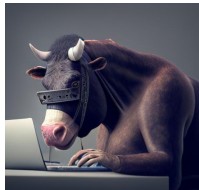

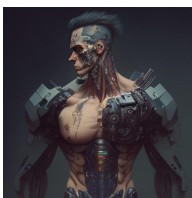

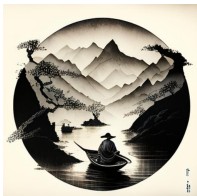

nature vs human nature, surreal, UHD, 8k, hyper details, rich colors, photograph.

an Emu, focused yet playful, ready for a competitive matchup, photorealistic quality with cartoon vibes.

A worker that looks like a mixture of cow and horse is working hard to type code.

Half human, half robot, repaired human, human flesh warrior, mech display, man in mech, cyberpunk.

a traveler navigating via a boat in countless mountains, Chinese ink painting.

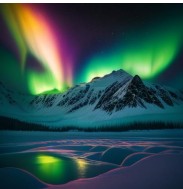

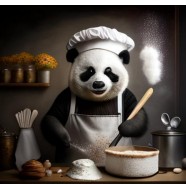

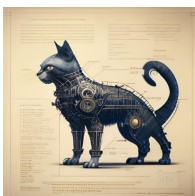

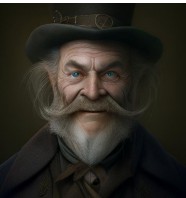

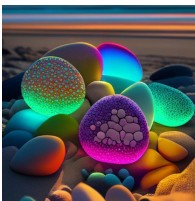

A realistic landscape shot of the Northern Lights dancing over a snowy mountain range in Iceland, with long exposure to capture the motion and vibrant colors.

A high contrast portrait of a very happy fuzzy panda dressed as a chef in a high end kitchen making dough. There is a painting of flowers on the wall behind him.

poster of a mechanical cat, technical Schematics viewed from front and side view on light white blueprint paper, illustration drafting style, illustration, typography, conceptual art, dark fantasy steampunk, cinematic, dark fantasy.

19th century Scottish wizard with a mysterious smile and a piercing gaze, enigmatic, photorealistic, incredibly detailed, sharpness, detail, cinematic lighting

several brightly colored rocks on a colorful beach, in the style of luminous spheres, 3840x2160, emek golan, translucent color, 32k uhd, toyen, captivating

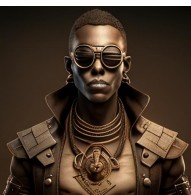

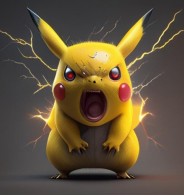

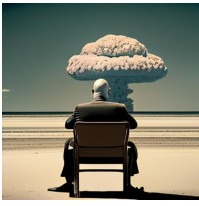

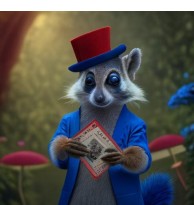

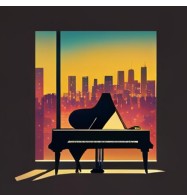

A photograph of a portrait of a statue of a pharaoh wearing steampunk glasses, white t-shirt and leather jacket.

A Pikachu with an angry expression and red eyes, with lightning around it, hyper realistic style.

Oppenheimer sits on the beach on a chair, watching a nuclear exposition with a huge mushroom cloud, 120mm.

A 4k dslr image of a lemur wearing a red magician hat and a blue coat performing magic tricks with cards in a garden.

A silhouette of a grand piano overlooking a dusky cityscape viewed from a top-floor penthouse, rendered in the bold and vivid style of a vintage travel poster.

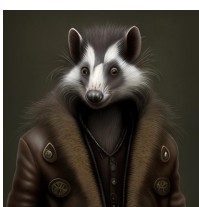

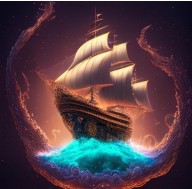

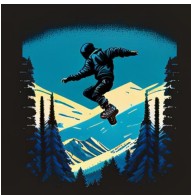

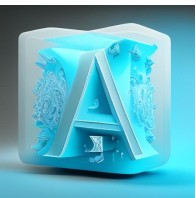

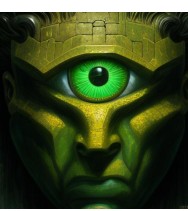

In a fantastical setting, a highly detailed furry humanoid skunk with piercing eyes confidently poses in a medium shot, wearing an animal hide jacket. The artist has masterfully rendered the character in digital art, capturing the intricate details of fur and clothing texture.

Pirate ship trapped in a cosmic maelstrom nebula, rendered in cosmic beach whirlpool engine, volumetric lighting, spectacular, ambient lights, light pollution, cinematic atmosphere, art nouveau style, illustration art artwork by SenseiJaye, intricate detail.

Art style of a snowboarder in mid-air performs a trick on a black rail, wearing a blue sweatshirt and black pants, with arms outstretched. The serene snowy landscape background, dotted with trees, complements the scene. The low-angle perspective emphasizes the trick's height and skill.

Design a letter A, 3D stereoscopic Ice material Interior light blue Conceptual product design Futuristic Blind box toy Handcrafted Exquisite 3D effect Full body display Ultra-high precision.

An ancient stone Colossus with eye, Stephan Martinière, dark yellow and light emerald, color zone painting, Denis Sarazhin, dark emerald and silver, robotic expressionism, high detail.

Figure 8: **Text-conditional 512×512 image generation samples with long prompts.** The results are generated by BiGR-XL-d24.

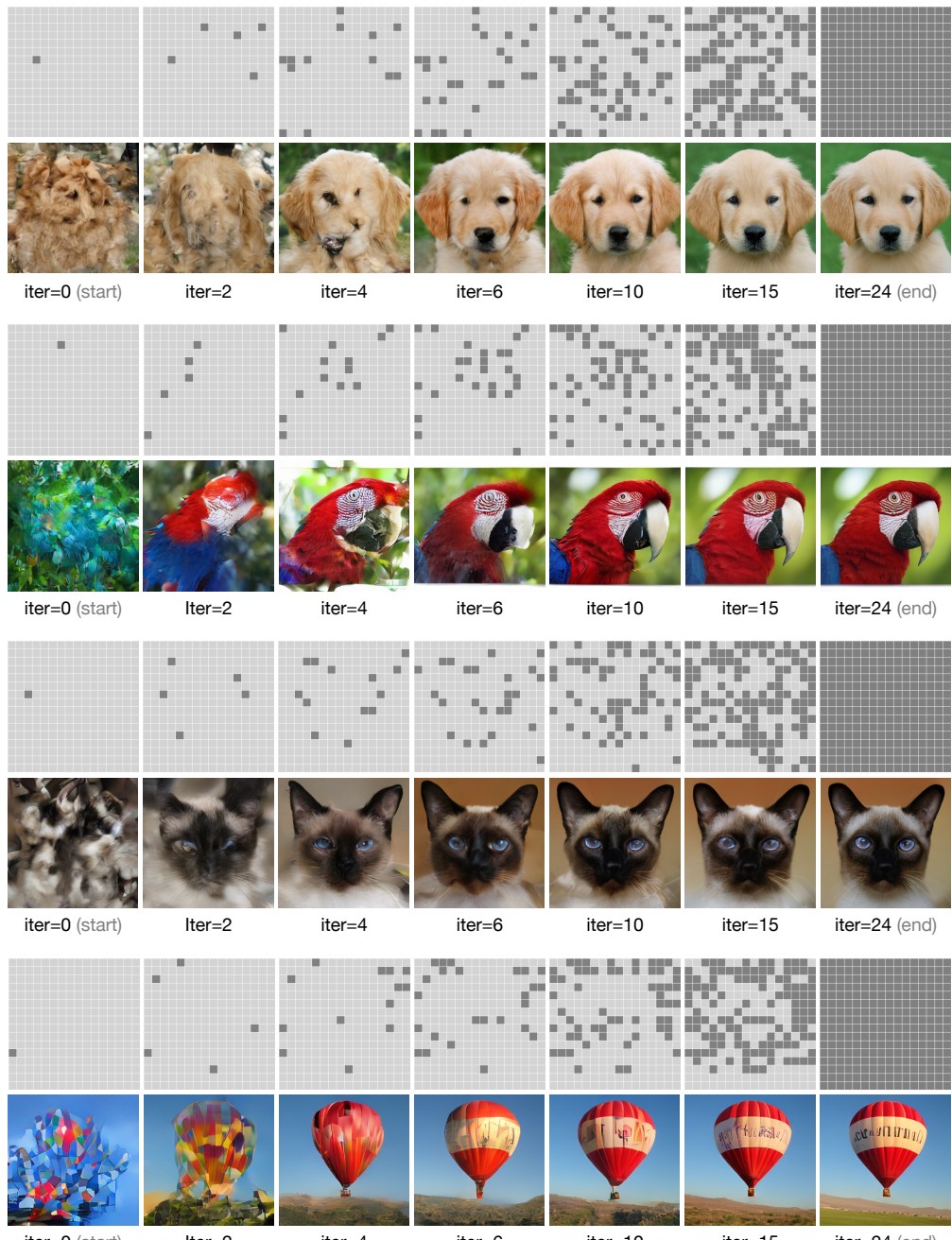

Figure 9: **Visualization of the entropy-ordered sampling process.** For each generation sample, we present the illustration of the unmasked tokens at the top and the corresponding generated images at the bottom for each sampling iteration. At each iteration, the masked positions are filled with the output tokens from the current iteration. Here, we use BiGR-XL-d24, setting the total number of sampling iterations to 25.

## D    FAILURE CASES

We show some failure cases of our generation results in Fig. 10. It is challenging to generate high-quality, authentic human faces and fingers, clear and accurate numbers and signs, and highly complex scenes with many objects. These issues are common in conditional generative models, arising from the significant visual quality differences across categories in the ImageNet-1K training dataset and the autoencoder's limited ability to reconstruct complex details.

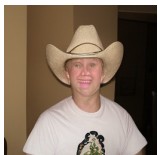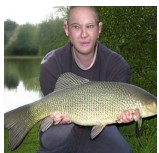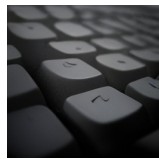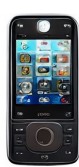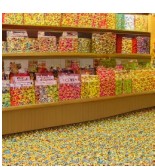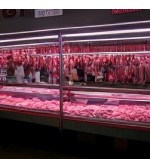

Figure 10: **Failure cases** in generating high-quality human faces and fingers, clear numbers and signs, and complex scenes.

## E  ADDITIONAL SYSTEM-LEVEL COMPARISON

We provide a more comprehensive comparison of different leading models. We compare generative performance in Tab. 9 and discriminative performance in Tab. 10.

## F  ADDITIONAL GENERATED SAMPLES

We provide additional 512×512 samples and 256×256 samples generated by our model in Fig. 11. We also include uncurated generated samples from various classes in Fig. 12 to 23.

## G  ETHICS STATEMENT

We recognize the ethical risks of image generation, such as potential misuse for harmful content. Our research aims to promote positive uses like creativity and education, with a commitment to responsible application. Safeguards and continuous ethical oversight are strongly encouraged.

Table 9: **Model comparison of generative performance on ImageNet-1K.** Metrics include Frechet inception distance (FID), inception score (IS), precision (Pre.) and recall (Rec.). All models are tested on 256×256 ImageNet-1K benchmark. The suffix "-re" denotes the use of rejection sampling.

| Type | Model | #Params. | FID↓ | IS↑ | Pre.↑ | Rec.↑ |
|---|---|---|---|---|---|---|
| **GAN** | BigGAN (Brock, 2018) | 112M | 6.95 | 224.5 | 0.89 | 0.38 |
| | GigaGAN (Kang et al., 2023) | 569M | 3.45 | 225.5 | 0.84 | 0.61 |
| | StyleGanXL (Sauer et al., 2022) | 166M | 2.30 | 265.1 | 0.78 | 0.53 |
| **Diffusion** | LDM-4 (Rombach et al., 2022) | 400M | 3.60 | 247.7 | - | - |
| | DiT-L/2 (Peebles & Xie, 2023) | 458M | 5.02 | 167.2 | 0.75 | 0.57 |
| | DiT-XL/2 | 675M | 2.27 | 278.2 | 0.83 | 0.57 |
| | SiT-XL/2 (ODE) (Ma et al., 2024) | 675M | 2.15 | 258.1 | 0.81 | 0.60 |
| | SiT-XL/2 (SDE) | 675M | 2.06 | 277.5 | 0.83 | 0.59 |
| **Mask.** | MaskGIT (Chang et al., 2022) | 227M | 6.18 | 182.1 | 0.8 | 0.51 |
| | MaskGIT-re | 227M | 4.02 | 355.6 | - | - |
| **AR** | VQGAN (Esser et al., 2021) | 227M | 18.65 | 80.4 | 0.78 | 0.26 |
| | VQGAN | 1.4B | 15.78 | 74.3 | - | - |
| | VQGAN-re | 1.4B | 5.20 | 280.3 | - | - |
| | ViT-VQGAN (Yu et al., 2022a) | 1.7B | 4.17 | 175.1 | - | - |
| | ViT-VQGAN-re | 1.7B | 3.04 | 227.4 | - | - |
| | RQTran. (Lee et al., 2022) | 3.8B | 7.55 | 134.0 | - | - |
| | RQTran.-re | 3.8B | 3.80 | 323.7 | - | - |
| **VAR** | VAR-d16 (Tian et al., 2024) | 310M | 3.30 | 274.4 | 0.84 | 0.51 |
| | VAR-d20 | 600M | 2.57 | 302.6 | 0.83 | 0.56 |
| | VAR-d24 | 1.0B | 2.09 | 312.9 | 0.82 | 0.59 |
| | VAR-d30 | 2.0B | 1.92 | 323.1 | 0.82 | 0.59 |
| **MAR** | MAR-B (Li et al., 2024) | 208M | 2.31 | 281.7 | 0.82 | 0.57 |
| | MAR-L | 479M | 1.78 | 296.0 | 0.81 | 0.60 |
| | MAR-H | 943M | 1.55 | 303.7 | 0.81 | 0.62 |
| **AR** | LlamaGen-B (Sun et al., 2024) | 111M | 5.46 | 193.6 | 0.83 | 0.45 |
| | LlamaGen-L | 343M | 3.81 | 248.3 | 0.83 | 0.52 |
| | LlamaGen-XL | 775M | 3.39 | 227.1 | 0.81 | 0.54 |
| | LlamaGen-XXL | 1.4B | 3.09 | 253.6 | 0.83 | 0.53 |
| | LlamaGen-3B | 3.1B | 3.05 | 222.3 | 0.80 | 0.58 |
| **Ours** | BiGR-L-d24 | 336M | 2.71 | 275.7 | 0.84 | 0.53 |
| | BiGR-XL-d24 | 799M | 2.49 | 278.8 | 0.84 | 0.55 |
| | BiGR-XXL-d24 | 1.5B | 2.36 | 277.2 | 0.83 | 0.55 |

Table 10: **Linear-probe top-1 accuracy on ImageNet-1K.** MIM denotes masked image modeling. †: our evaluation results.

| | Method | #Tokens | Params | ACC1↑ |
|---|---|---|---|---|
| **Contrastive methods** | CPC v2 (Henaff, 2020) | - | 303M | 71.5 |
| | MoCo (He et al., 2020) | - | 375M | 68.6 |
| | SimCLR (Chen et al., 2020b) | - | 375M | 76.5 |
| | SwAV (Caron et al., 2020) | - | 93M | 75.3 |
| | DINO (Caron et al., 2021) | - | 85M | 75.3 |
| | BYOL (Grill et al., 2020) | - | 375M | 78.6 |
| | CAE (Chen et al., 2024c) | - | 304M | 78.1 |
| | CMAE (Huang et al., 2023) | - | 86M | 73.9 |
| **MIM** | iBOT (Zhou et al., 2022) | - | 304M | 81.0 |
| | BEiT (Bao et al., 2022) | 16×16 | 307M | 73.5 |
| | MAE (He et al., 2022) | 14×14 | 304M | 75.8 |
| | MAGE (Li et al., 2023a) | 16×16 | 328M | 78.9 |
| **Generative methods** | BiGAN Donahue et al. (2017) | - | 138M | 31.0 |
| | BigBiGAN (Donahue & Simonyan, 2019) | - | 86M | 56.6 |
| | BigBiGAN | - | 344M | 61.3 |
| | iGPT-L (Chen et al., 2020a) | 32×32 | 1.4B | 60.3 |
| | iGPT-L | 48×48 | 1.4B | 65.2 |
| | ViT-VQGAN-B (Yu et al., 2022a) | 32×32 | 650M | 65.1 |
| | ViT-VQGAN-L | 32×32 | 1.7B | 73.2 |
| | RCG (Li et al., 2023b) | 16×16 | 304M | 77.6 |
| | *l*-DAE (Chen et al., 2024d) | - | 304M | 75.0 |
| **Cond. gen.** | LlamaGen-L† (Sun et al., 2024) | 16×16 | 343M | 40.5 |
| | MAR-B† (Li et al., 2024) | 16×16 | 208M | 57.9 |
| | MAR-L† | 16×16 | 479M | 59.1 |
| | MAR-H† | 16×16 | 943M | 60.0 |
| | BiGR-L-d20 (Ours) | 16×16 | 336M | 67.5 |
| | BiGR-XL-d32 (Ours) | 16×16 | 799M | 69.8 |

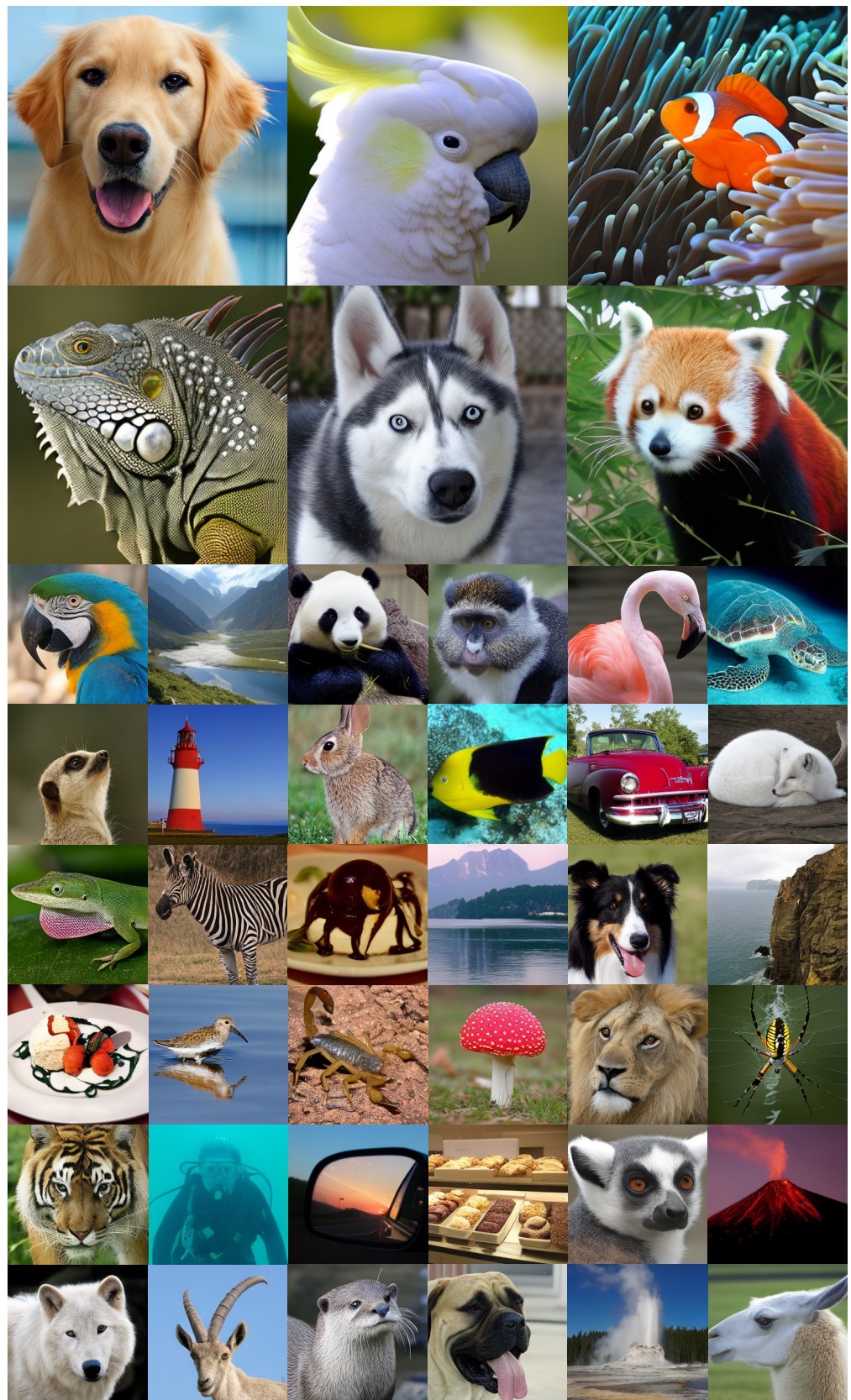

Figure 11: **Additional generated 256×256 and 512×512 samples.**

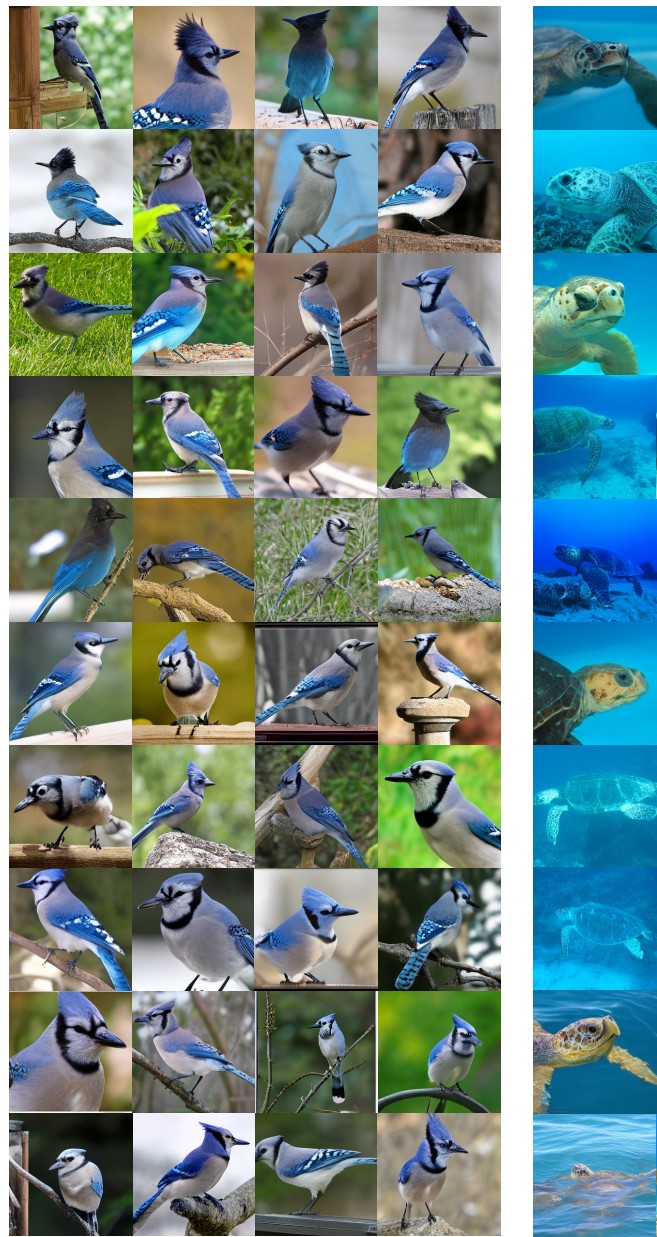 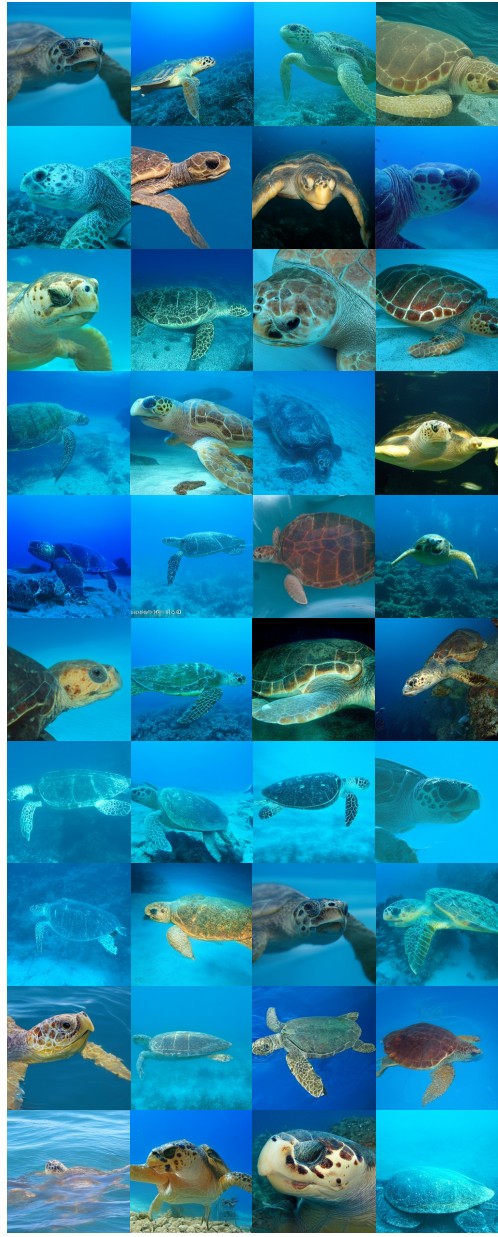

Figure 12: **Uncurated** 256×256 samples. Model: BiGR-XXL-d32 Class label = "Jay" (17)

Figure 13: **Uncurated** 256×256 samples. Model: BiGR-XXL-d32 Class label = "Loggerhead, loggerhead turtle, Caretta caretta" (33)

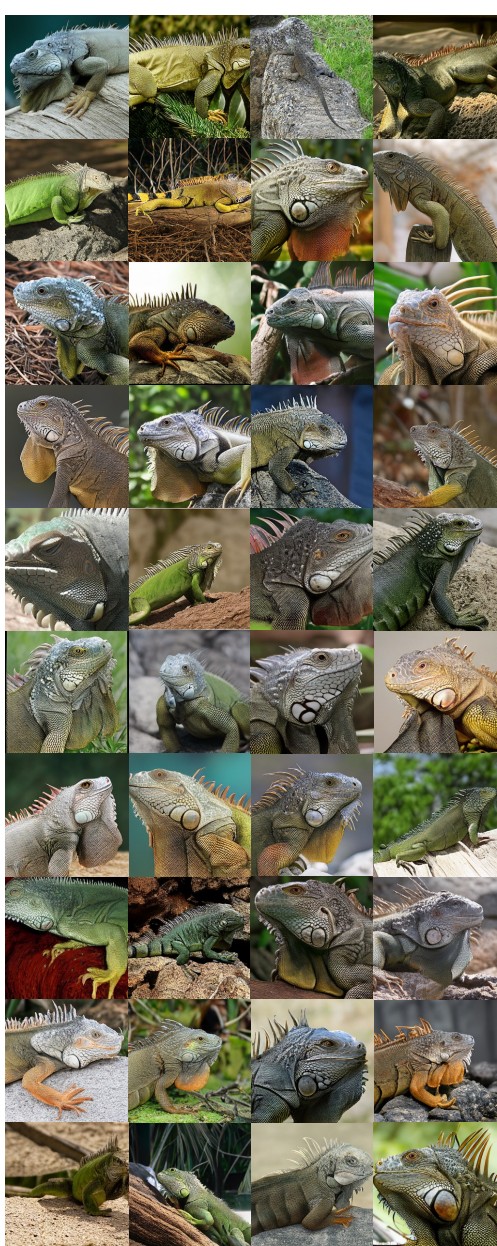 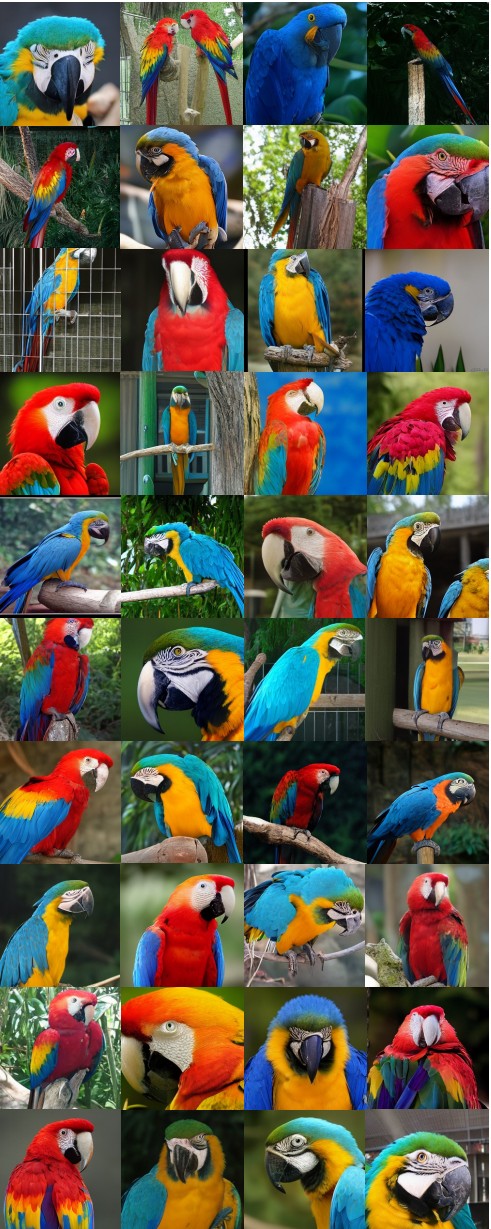

Figure 14: **Uncurated** 256×256 samples. Model: BiGR-XXL-d32 Class label = "Common iguana, Iguana, Iguana iguana" (39)

Figure 15: **Uncurated** 256×256 samples. Model: BiGR-XXL-d32 Class label = "Macaw" (88)

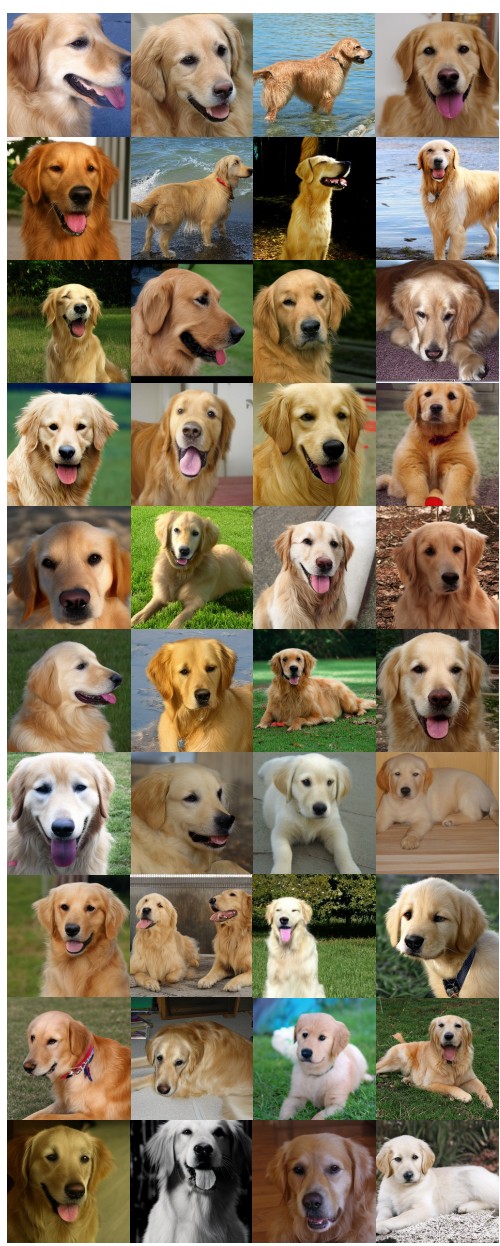 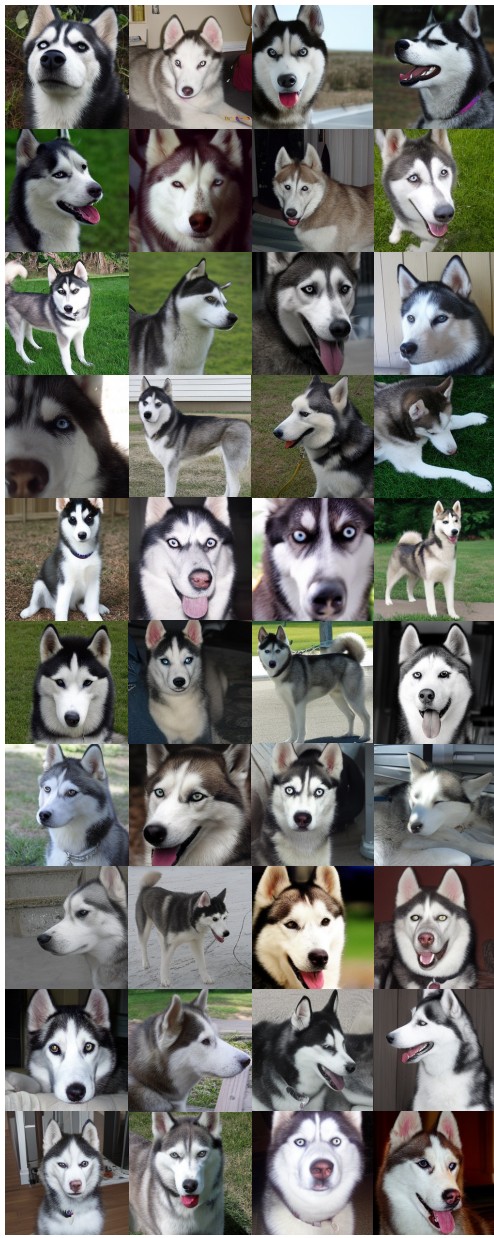

Figure 16: **Uncurated** 256×256 samples.
Model: BiGR-XXL-d32
Class label = "Golden retriever" (207)

Figure 17: **Uncurated** 256×256 samples.
Model: BiGR-XXL-d32
Class label = "Siberian husky" (250)

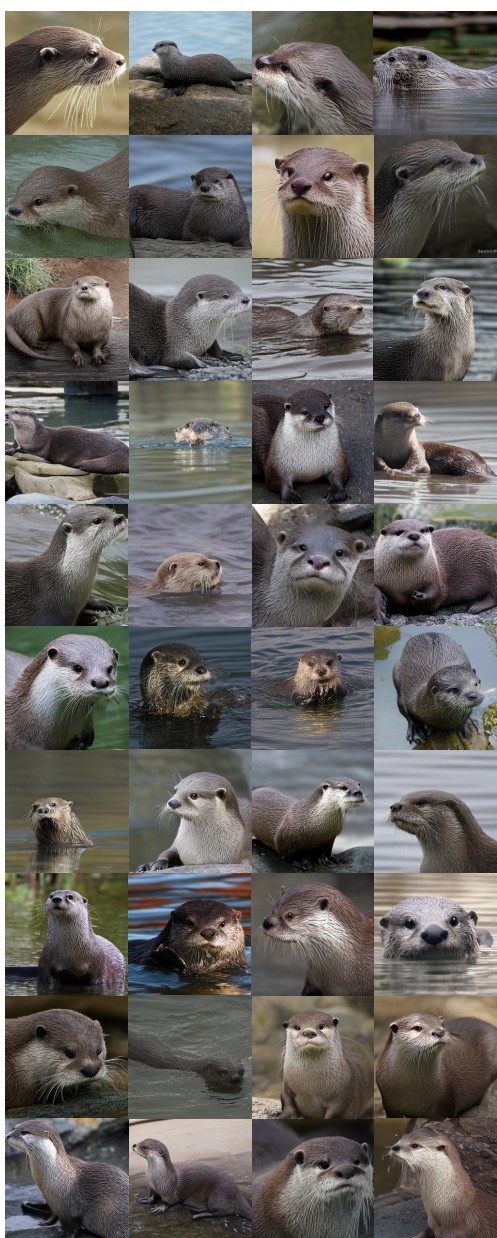 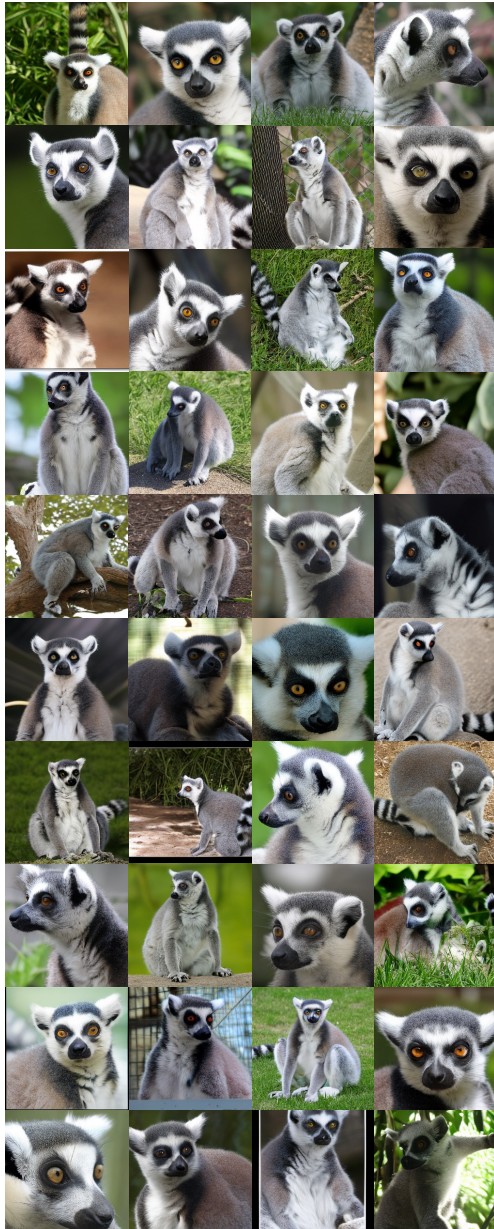

Figure 18: **Uncurated** 256×256 samples. Model: BiGR-XXL-d32 Class label = "Otter" (360)

Figure 19: **Uncurated** 256×256 samples. Model: BiGR-XXL-d32 Class label = "Madagascar cat, ring-tailed lemur, Lemur catta" (383)

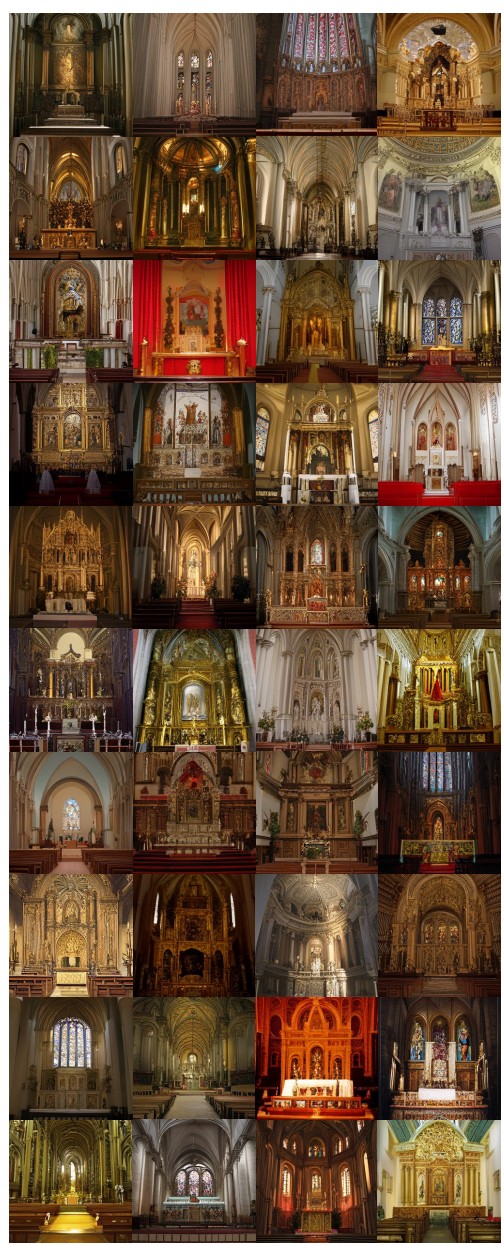 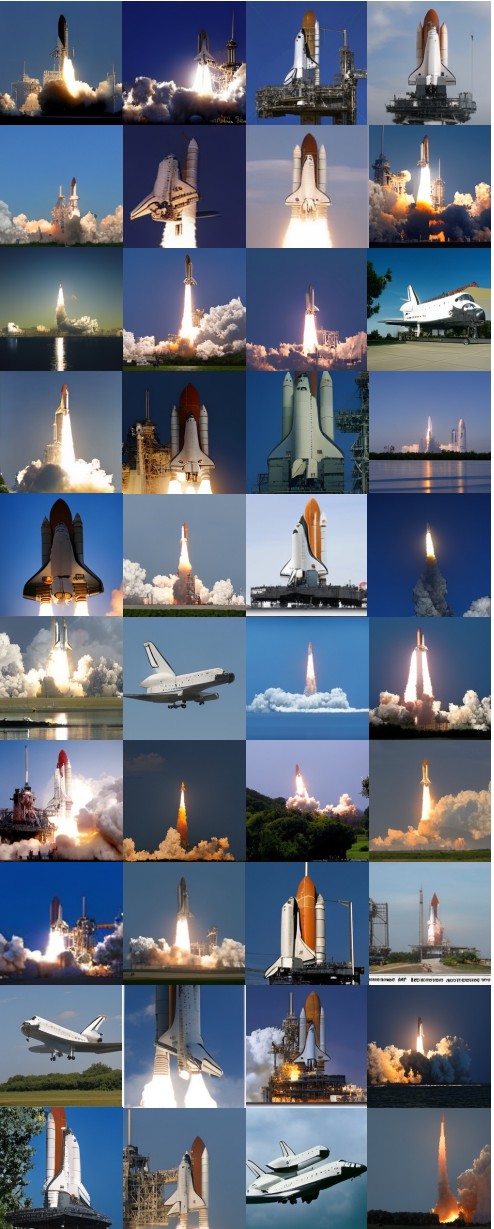

Figure 20: **Uncurated** 256×256 samples.
Model: BiGR-XXL-d32
Class label = "Altar" (406)

Figure 21: **Uncurated** 256×256 samples.
Model: BiGR-XXL-d32
Class label = "Space Shuttle" (812)

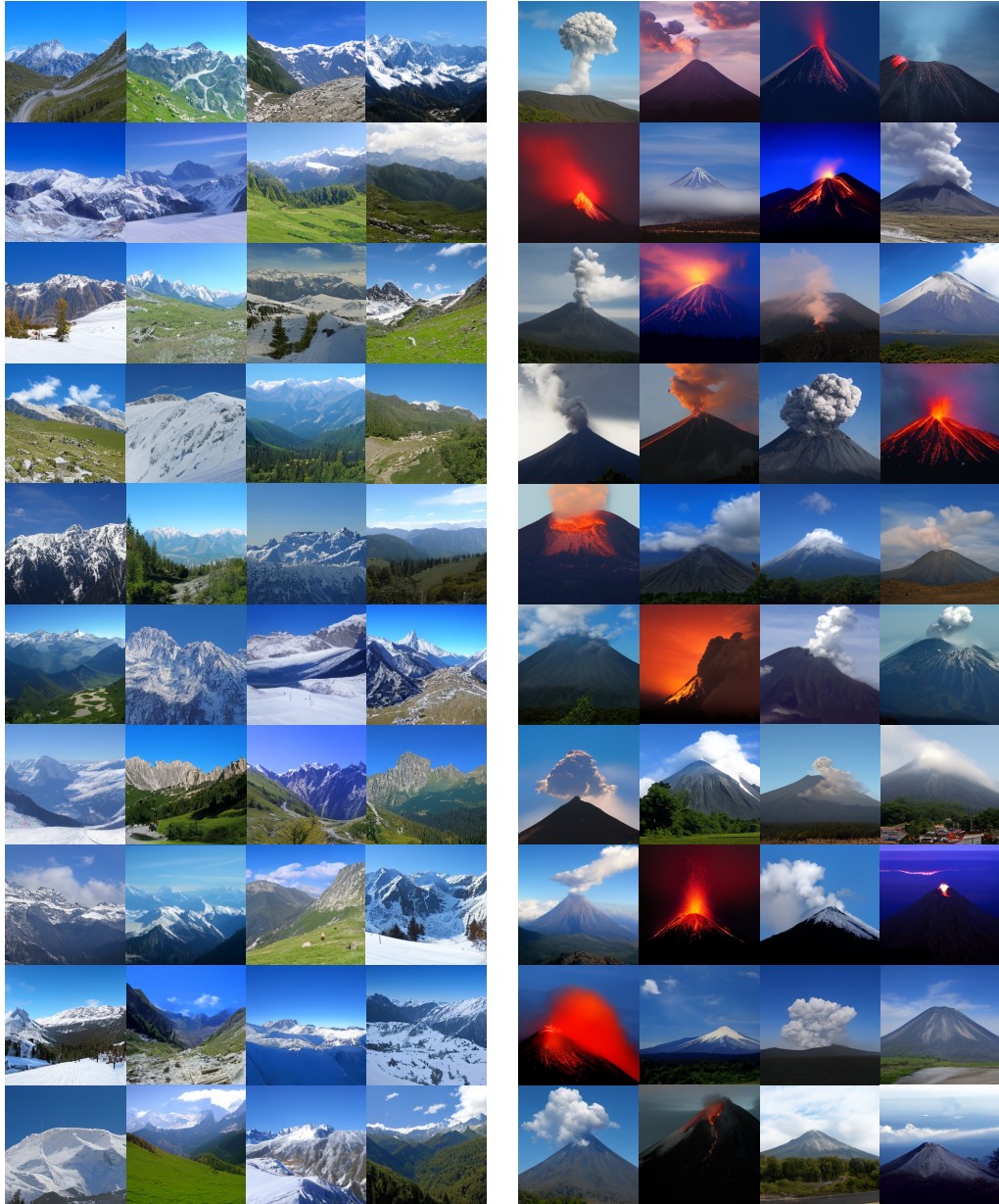

Figure 22: **Uncurated** 256×256 samples.
Model: BiGR-XXL-d32
Class label = "Alp" (970)

Figure 23: **Uncurated** 256×256 samples.
Model: BiGR-XXL-d32
Class label = "Volcano" (980)

