# OpenReview forum: "BiGR: Harnessing Binary Latent Codes for Image Generation and Improved Visual Representation Capabilities"
_ICLR.cc/2025/Conference — ICLR 2025 Poster_

### Official Review · Reviewer_FMho · 2024-11-02

**Soundness:** 2
**Presentation:** 2
**Contribution:** 2
**Rating:** 6
**Confidence:** 5

**Summary:**

The authors propose a language model based image generation/discrimination model. Using binary latent code autoencoder, the model can learn binary codes from the image representation. Llama is originally a decoder-only model but this method use it as encoder-only model. The generation of an image is conducted by sampling from the Bernoulli distribution of outputs of the model.

**Strengths:**

- The method is easy to follow and the paper is easy to read
- The architecture of this model seems to work very well on specific tasks such as inpainting, outpainting.

**Weaknesses:**

- The biggest problem of this approach is that it is unable to conduct text-to-image generation. Early stage of diffusion models were constrained to class-conditional generation but now it is hard to find models that is unable to do t2i generation. Even LlamaGen can receive various types of condition (especially text condition) since it is a decoder-only model.
- I think that's why binary latent code has been enough to encode image representation. Even VQ-VAE inevitably suffers from loss of information because the latent variable is not continuous. But as the problem setting of this paper is limited to class conditional image generation, the amount of information is not large enough to see the malfunction of the binary code.
- Also, I think it is not fair to directly compare with LlamaGen since it is designed to handle multiple modalities, not focusing on image generation. And also with an appropriate prompt, LlamaGen is able to conduct image discrimination task as well.
- In conclusion, limiting the scope of the problem enabled the binary code to work well.

**Questions:**

- Comments in Weaknesses should be resolved.

---

> ### Author Response · Authors · 2024-11-19
> **Response to Reviewer FMho**
>
> Thank you for your valuable reviews! We address the concerns below:
>
> ---
>
> ### W1: Text-to-image generation
>
> The reviewer’s main concern is that BiGR is trained only for class-conditional generation and not for text-to-image generation.
>
> We strongly agree that text-to-image generation is an important application to image generation models, while we also believe that studying the core class-conditional image generation model is **of fundamental importance**. The success of existing text-conditional image generation models is largely built on their class-conditional counterparts. This remains a crucial area of research, as evidenced by recent works such as SiT [1], VAR [2], and MAR [3].
>
> Meanwhile, to comprehensively answer the reviewer’s query, we also enable BiGR to **perform text-to-image generation** by training it on a large-scale image-caption dataset. We train our XL-d24 model using 4M JourneyDB dataset [4] on 32 A800s for 62 hours. We reported our text-to-image generation results in Appendix B of the revised paper.
>
> **The results reveal the strong potential of BiGR in text-to-image generation.**
>
> Note that, due to time constraints, our training data (4M), model size (< 800M), and training duration (< 3 days) are relatively limited, and we only train a 256$\times$256 T2I model, which has a relatively low image resolution.
> We believe that scaling up training data and model size, along with exploring advanced techniques such as incorporating higher resolutions, fine-tuning autoencoders, and applying multi-stage training, can further enhance BiGR's T2I performance. We will explore this in future research.
>
> ### W2: Effectiveness of binary latent code
>
> The success of text-to-image generation demonstrates that the binary latent code can **encode sufficient image information without malfunction**.
>
> We agree with the reviewer’s viewpoint that the image tokenizer can introduce information loss, which is *inevitable* for all image tokenizers due to **image compression**.
>
> The reconstruction FID (rFID) metric is commonly used to evaluate the performance of image tokenizers. We compare our binary autoencoder with the widely used VQVAE employed in LlamaGen. The results are shown below:
>
> | Tokenizer | dim  | tokens        | rFID |
> | --------- | :----: | :-------------: | :----: |
> | VQVAE  | -    | 16$\times$16  | 2.19 |
> | B-AE      | 16   | 16$\times$16  | 3.32 |
> | B-AE      | 20   | 16$\times$16  | 2.25 |
> | B-AE      | 24   | 16$\times$16  | 1.78 |
> | B-AE      | 32   | 16$\times$16  | **1.69** |
>
> These comparable rFID results also demonstrate that the binary code can effectively encode sufficient image information.
>
> The rFIDs of B-AE have been plotted in Figure 4 (left) in the paper.
>
> Besides, autoencoders that produce binary codes have also proven to be effective in works such as Binary Latent Diffusion [5] and MAGVIT-v2 [6].
>
> ### W3: Fair comparison with LlamaGen
>
> We would like to respectfully clarify some of the reviewer's descriptions about LlamaGen:
>
> 1. LlamaGen is an **image generation model** that handles two modality conditions: class and text. It can perform class-conditional and text-conditional generation, with these models trained separately.
>
> 2. Since LlamaGen is *solely* an image generation model, it **cannot** perform image discrimination tasks through prompting.
>
> We kindly ask the reviewer to refer to the original LlamaGen paper [7] for more details, which may help verify the points discussed above and address the reviewer’s concerns.
>
> To this end, the comparison between BiGR and LlamaGen is **fair**. We list the tasks that LlamaGen and BiGR can do:
> - **LlamaGen**: class-conditional image generation, text-to-image generation
> - **BiGR**: class-conditional image generation, text-to-image generation, improved discrimination, zero-shot applications (e.g., inpainting, outpainting, editing, interpolation, and enrichment).
>
> Thus, BiGR can perform more tasks than LlamaGen.
>
> ### Conclusion
>
> All of the above evidence sufficiently demonstrates that the binary code works well across a broad scope.
>
> ---
>
> We greatly appreciate the reviewer's suggestions, which help strengthen our paper with the addition of the text-to-image generation results.
>
> **If the reviewer has additional concerns, we would be happy to discuss them!**
>
> [1] Ma et al. SiT: Exploring Flow and Diffusion-based Generative Models with Scalable Interpolant Transformers. ECCV 2024.
> [2] Tian et al. Visual Autoregressive Modeling: Scalable Image Generation via Next-Scale Prediction. NeurIPS 2024.
> [3] Li et al. Autoregressive Image Generation without Vector Quantization. NeurIPS 2024.
> [4] Sun et al. JourneyDB: A benchmark for generative image understanding. NeurIPS 2023.
> [5] Wang et al. Binary Latent Diffusion. CVPR 2023.
> [6] Yu et al. Language Model Beats Diffusion: Tokenizer is key to visual generation. ICLR 2024.
> [7] Sun et al. Autoregressive Model Beats Diffusion: Llama for Scalable Image Generation. https://arxiv.org/abs/2406.06525

---

> ### Comment · Reviewer_FMho · 2024-11-19
>
> - I now agree with the philosophy of the authors that this is about fundamental research.
> - The authors have presented text-to-image generation with good performance. At the final version, I recommend to add results with further training for the best.
> - I appreciate for correcting me about LlamaGen.
> - Considering all revision conducted on this paper, I would like to raise my score to '6: marginally above the acceptance threshold'.

---

> > ### Author Response · Authors · 2024-11-19
> > **Thank you!**
> >
> > Thank you for your reply and recognition of our work! We are happy your concerns are addressed.
> >
> > We will continue to improve our results with further training of our T2I model!

---

### Official Review · Reviewer_bv3t · 2024-11-02

**Soundness:** 3
**Presentation:** 3
**Contribution:** 3
**Rating:** 6
**Confidence:** 3

**Summary:**

The paper introduces BiGR, a novel conditional image generation model that unifies generative and discriminative tasks within a single framework. This model is notable for several key advantages:

Uniformity: BiGR is the first model to integrate both generative and discriminative tasks, leveraging compact binary latent codes to achieve strong performance in both areas. This unification allows BiGR to handle tasks that typically require separate models.

Efficiency: The model is designed to generate images quickly, making it more efficient than existing models. This efficiency is achieved without compromising the quality of the generated images.

Flexibility and Scalability: BiGR is adaptable to various tasks, including zero-shot generalized tasks, showcasing its potential for a wide range of applications. The model's scalability is demonstrated through its performance across different model sizes and configurations.

Performance: Extensive experiments show that BiGR delivers decent performance in terms of generation quality and linear separability. The model's performance is evaluated using metrics like FID (Fréchet Inception Distance), and it is shown to perform well compared to other models like LlamaGen.

Inference Hyperparameters: The paper discusses the impact of hyperparameters such as the number of sampling iterations and diffusion timesteps on the model's performance. It is noted that larger models tend to achieve lower FID values, but with increased sample time, and that optimal performance varies with model size.

Comparison with Other Models: BiGR is compared against other models, including LlamaGen, across different settings involving tokenizers, training objectives, and modeling types. The paper highlights that while the unconditional version of the model shows better representation capabilities, the conditional version excels in generative tasks.

Overall, BiGR represents a significant advancement in the field of image generation by combining generative and discriminative capabilities in a single, efficient model, with promising applications for future research and development.

**Strengths:**

Please refer to the summary.

**Weaknesses:**

Lack of Comprehensive Benchmarking: While the paper compares its model against LlamaGen and a few other settings, the scope of comparison is limited. The paper could benefit from a more extensive benchmarking against a wider range of state-of-the-art models to better establish its relative performance.

Sampling Strategy Issues: The paper mentions a "nan" issue in the sampling strategy due to logarithmic operations. Although a workaround is provided, this indicates potential instability in the model's implementation. A more robust solution to this problem would enhance the reliability of the model.

Limited Exploration of Model Configurations: The paper primarily focuses on a few configurations (S0, S1, S2, S3) and does not explore a broader range of hyperparameters or architectural variations. This limits the understanding of the model's capabilities and its adaptability to different tasks or datasets.

Evaluation Metrics: The paper emphasizes generative performance but does not provide a detailed analysis of other important aspects such as scalability, robustness, or efficiency. Including these metrics would provide a more holistic view of the model's strengths and weaknesses.

Assumptions and Limitations: The paper acknowledges that surpassing state-of-the-art models across all metrics is not the goal, but it does not clearly outline the specific scenarios or applications where the proposed model excels. A clearer articulation of the model's intended use cases and limitations would help in understanding its practical applicability.

Theoretical Justification: While empirical results are presented, the paper could strengthen its theoretical foundation by providing more in-depth explanations or proofs of why certain design choices, such as the non-deterministic binary transcoder, lead to better performance.

**Questions:**

Please refer to the weakness.

---

> ### Author Response · Authors · 2024-11-19
> **Response to Reviewer bv3t**
>
> Thank you for your review! We address the concerns below:
>
> ---
>
> ### W1: Benchmark
>
> The benchmarks used in this paper (image generation and linear probing on the ImageNet-1K 256$\times$256 validation split) are widely adopted in the fields of generation and representation learning. The evaluation metrics we report, including FID, Inception Score (IS), sFID, Precision (Pre.), Recall (Rec.), and linear-probe accuracy, provide a comprehensive assessment of the model performance.
>
> We comprehensively compare BiGR with the state-of-the-art generative models in Tables 5 and 9, and the state-of-the-art discriminative models in Tables 6 and 10. We also added the results of SiT [1] in Tables 5 and 9 of the revised paper.
>
> Despite this, we would like to re-emphasize that the goal of this work is to propose a uniform conditional generative model that can produce high-quality generations while maintaining strong representation capabilities. Therefore, surpassing state-of-the-art models across all metrics is not within the scope of this research.
>
> ### W2: NaN issue with logarithmic operation
>
> We use the alternative method $2 \times |p_k - 0.5|$ to compute confidence in sampling, as described in Appendix A (L750-754). This method functions *identically* to the original logarithmic operation but avoids the NaN issue. It is **robust**, and we did not encounter any problems in any of our experiments.
>
> ### W3: Model exploration
>
> ***Hyperparameters or architectural variations.*** We have explored different binary transcoders in Table 2, compared different sampling orders in Table 3, and tested varying sampling iterations and varying diffusion timesteps in Figure 3.
>
> Please see more discussions on hyperparameters in the **response to Reviewer jehB (W1)**. Due to the high training cost, it is infeasible to fully explore different architectural variations.
>
> ***Adaptability to different tasks or datasets.*** We have shown our model can perform multiple vision tasks, including inpainting, outpainting, editing, interpolation, and enrichment, as presented in Figure 6. We further adapt our model for text-to-image generation by training it on a large-scale image-caption dataset. We included the results in Appendix B of the revised paper. These results demonstrate the **adaptability** of our model.
>
> ### W4: Evaluation metrics
>
> 1. **Scalability**: We have discussed the metrics of FID-50K and linear-probe ACC across different model scales in Figure 4, validating the **scalability** of our model.
> 2. **Robustness**: The FID metric is evaluated with 50K randomly generated images, which demonstrates the **robustness** of the model. Additionally, our model can handle various zero-shot generalized applications, as shown in Figure 6, further indicating its **robustness** across different tasks.
> 3. **Efficiency**: We have compared models’ inference times in Tables 1 & 3 and Figure 3, which highlights the BiGR’s **efficiency**.
>
> ### W5: Application and limitation
>
> BiGR unifies conditional generation and discrimination within the same framework, **a scenario not previously explored**. BiGR is the *first* to demonstrate strong uniformity in both generation and representation capabilities for a conditional image generation model.
>
> Additionally, BiGR supports zero-shot applications, such as inpainting, outpainting, editing, interpolation, and enrichment, as demonstrated in Figure 6.
>
> Furthermore, we enable BiGR to perform text-to-image generation, a significant application for generative models.
>
> We have discussed our limitations, including **hyperparameter complexity** and **fixed sequence length**, in L537-539. Additional discussions of our limitations can be found in our **response to Reviewer jehB (W1&W2)**.
>
> ### W6: Theoretical justification
>
> A detailed theoretical justification of the Bernoulli diffusion forward and denoising processes (in our binary transcoder) is provided in [2]. The other components of our model are designed to be intuitive and straightforward. Hence, we do not provide theoretical justifications for them.
>
> We conjecture that the non-deterministic binary transcoder achieves better performance because it enhances the diversity of generated images, leading to improved metrics.
>
> ---
>
> **We are happy to discuss any further concerns the reviewer may have!**
>
> [1] Ma et al. SiT: Exploring Flow and Diffusion-based Generative Models with Scalable Interpolant Transformers. ECCV 2024.
> [2] Wang et al. Binary Latent Diffusion. CVPR 2023.

---

> > ### Author Response · Authors · 2024-11-21
> > **Follow-up**
> >
> > We hope our response has addressed your concerns! If you have any further questions, please feel free to discuss them with us.
> >
> > You can also find some exciting text-to-image generation results in our revised paper. Thank you!

---

> > > ### Comment · Reviewer_bv3t · 2024-11-21
> > >
> > > The author's response resolved my issue, and I have decided to maintain my rating.

---

> > > > ### Author Response · Authors · 2024-11-21
> > > > **Thank you!**
> > > >
> > > > Thank you for your reply! We are glad that we have addressed your concerns!

---

### Official Review · Reviewer_jehB · 2024-11-04

**Soundness:** 4
**Presentation:** 3
**Contribution:** 3
**Rating:** 8
**Confidence:** 3

**Summary:**

The paper introduces BiGR, a conditional image generation model that leverages compact binary latent codes to achieve both high-quality image generation and strong visual representation capabilities. BiGR integrates a binary tokenizer, a masked modeling mechanism, and a binary transcoder to generate binary codes, to achieve efficient generation through an entropy-ordered sampling strategy. The model's design allows it to perform favorably in both generative and discriminative tasks. BiGR demonstrates strong performance on generation metrics, e.g., FID-50k and representation tasks evaluated via linear-probe accuracy. Additionally, the proposed method demonstrates its versatility in applications including image editing and zero-shot generalization on several tasks.

**Strengths:**

- Unified Framework for Generation and Representation. The proposed method effectively combines generative and discriminative capabilities within a single model, demonstrating strong performance in both areas.
- Strong Experimental Validation. The model's performance is validated through extensive experiments, showing improvements over previous methods in generation quality and discriminative accuracy.
- Fast Inference Speed. The model’s entropy-ordered sampling strategy accelerates the generation process by iteratively unmasking tokens in a confidence-guided manner. This is significantly faster compared to autoregressive models.
- Various Applications. BiGR's ability to perform tasks such as inpainting, outpainting, and image enrichment in a zero-shot setting validates its flexibility and generalization capabilities.
- Extensive Ablations. The paper provides thorough ablation studies that detail the impact of various components and settings on the model's performance.
- Well written. The paper's motivation is clear and well-connected to the approach. Although some technical parts can be improved with more detail, the paper is well-written overall.
- The limitations are discussed in the paper.

**Weaknesses:**

- Hyperparameter Complexity. The proposed method relies on several hyperparameters for both training and inference, such as the CFG scale, Gumbel temperature, number of sampling iterations, and number of diffusion steps. This complexity increases the time and resources required for tuning. This is discussed in the limitation section of the paper.

- Fixed Sequence Length: The model’s architecture enforces a fixed sequence length during training, which restricts its flexibility to handle inputs of varying sizes. Generating images at different resolutions requires retraining the model with the new sequence length configuration. This is also discussed in the limitation section of the paper.

- The diffusion and denoising process is a bit confusing. It took me a while to figure out where the noise and denoising process is applied. Clarifying that the binary transcoder is the component responsible for denoising the noise introduced in the "Bernoulli diffusion" section would make the flow more understandable and easier to follow.

**Questions:**

1.	How does BiGR handle scenarios where binary latent codes introduce quantization artifacts?
2.	Does entropy order sampling prioritize representative features (attribute) of an object or class? Is there any relation in order of sampling and semantic characteristics?
3.	It would be insightful to include examples of failure cases.

---

> ### Author Response · Authors · 2024-11-19
> **Response to Reviewer jehB**
>
> Thank you for your effort and recognition of our work! We address the concerns below:
>
> ---
>
> As the reviewer acknowledged, we have discussed the two limitations in the paper. We would like to elaborate further to provide more insights.
>
> **W1: Hyperparameter complexity**
>
> BiGR is a novel model, so there is no prior experience to guide hyperparameter tuning. We have conducted extensive experiments to identify optimal hyperparameters. For each hyperparameter, we observe the following patterns:
>
> 1. **CFG Scale**: A larger CFG scale produces smoother images with clearer class features, while a smaller CFG scale enhances fine-grained details.
> 2. **Gumbel temperature**: A higher Gumbel temperature increases generation diversity but reduces quality, and vice versa.
> 3. **Number of sampling iterations**: 20–30 iterations generally perform well. Using 10 iterations speeds up generation but slightly reduces quality, while more than 30 iterations has minimal impact but slows down generation.
> 4. **Number of diffusion timesteps**: 100 steps typically yield good results. The broad range of 10–200 steps has only a marginal impact on performance.
>
> We added these observations in L707-718 in Appendix A of the revised paper. We believe that with our work and continued efforts from the community, BiGR’s hyperparameters can be further optimized.
>
>  **W2: Fixed Sequence Length**
>
> This limitation mainly arises from the need for the binary autoencoder to be retrained for different sequence lengths, requiring BiGR to be retrained accordingly. However, since the transformer architecture can handle varying sequence lengths, we can initialize the training for longer sequences (e.g., 32×32=1024) using a BiGR model pre-trained on shorter sequences (e.g., 16×16=256). This makes retraining more flexible.
>
> **W3: Clarification for better flow and easy understanding**
>
> Thank you for this valuable suggestion!
>
> We added the clarification that the binary transcoder component is responsible for noise denoising in L207-208 of Bernoulli diffusion paragraph in Sec. 3.1 of the revised paper.
>
> **Q1: Quantization artifact**
>
> The quantization artifact inherently comes with image autoencoders. It is also introduced by VQVAE/VQGAN.
>
> BiGR does not specially handle quantization artifacts but just follows and predicts the binary latent codes produced by the binary autoencoder. The potential issue of quantization artifacts is mitigated by training on large-scale data, as demonstrated by the generated results.
>
> We provide the reconstruction FID (rFID) of our binary autoencoders, compared to the rFID of VQVAE used in LlamaGen, in the **response to Reviewer FMho (W2)**. This reveals that our binary auto-encoder with code dimensions greater than 20 has lower quantization artifacts compared to VQVAE.
>
> **Q2: Priority of the entropy order**
>
> Interesting question! To explore this further, we visualize the generated results at different iterations during entropy-ordered sampling. We added this experiment in Appendix C of the revised paper, where the entropy-ordered sampling process is visualized in Figure 9.
>
> We observe that early iterations capture class-level characteristics, while subsequent iterations generate finer object-related details. In the final stages, visual quality steadily improves.
>
> **Q3: Failure cases**
>
> We added examples of failure cases in Appendix D of the revised paper.
>
> ---
>
> **If the reviewer has any further questions, feel free to discuss them with us!**

---

> > ### Author Response · Authors · 2024-11-21
> > **Follow-up**
> >
> > Thank you again for recognizing our work!
> >
> > Please don't hesitate to reach out with any questions or for further discussions. Additionally, we have included text-to-image generation results in the revised paper, which you might find interesting!

---

### Official Review · Reviewer_nzvc · 2024-11-04

**Soundness:** 2
**Presentation:** 2
**Contribution:** 2
**Rating:** 6
**Confidence:** 4

**Summary:**

BiGR is a novel conditional image generation model that uses compact binary latent codes to enhance both generative and representation capabilities.​ It unifies generative and discriminative tasks within the same framework, featuring a binary tokenizer, a masked modeling mechanism, and a binary transcoder for binary code prediction. BiGR introduces an entropy-ordered sampling method for efficient image generation and demonstrates superior performance in generation quality and representation capabilities.

**Strengths:**

1. Paper is clear and well-written.
2. The binary latent idea is new regarding Image Generation through LLMs.

**Weaknesses:**

- While the idea introduced is novel, it is hard for me to reason why the design choices used in the paper are leading to improving representation capabilities. It would be great if authors shed light on this much more.

- The idea of the diffusion process in Binary seems interesting, however the motivation of why it should improve the overall results could be clearer.

- The authors claim that they have replaced causal attention with bi-directional attention. I need help understanding how this can be done at the inference stage and what fine-tuning was done to make it work.

- The LlamaGen paper reports better results for the ImageNet (256x256). So could the authors please clarify the discrepancy in the results reported?

**Questions:**

- The SiT architecture reports improved results [1]; could authors clarify more about the SotA claim?

- Could the authors please fix the citation format at L196 by using \citet{}?

[1] Ma et al. ECCV 2024, SiT: Exploring Flow and Diffusion-based Generative Models with Scalable Interpolant Transformers

---

> ### Author Response · Authors · 2024-11-19
> **Response to Reviewer nzvc (part 1/2)**
>
> Thank you for your review! We address the concerns below:
>
> ---
>
> ### W1: Reasons behind the improved representation capabilities
>
> We attribute the improvement largely to three key design elements of our model:
>
> 1. **Masked modeling**: The masking mechanism uses the bidirectional attention, allowing all patch tokens to communicate with each other. This greatly benefits the integration of the global visual information, which enhances global feature representations across all tokens.
> 2. **Binary diffusion objective**: Unlike LLM-style implementations, such as LlamaGen, which project the transformer output into categorical logits, our binary transcoder predicts Bernoulli distributions conditioned on the intermediate transformer feature ($h$).
> This approach allows the feature to reconstruct element-wise binary codes, rather than being constrained to learn from a fixed codebook. As a result, the feature has the potential to capture richer information by avoiding the limitations of learning within a categorical space.
>
> These two designs are empirically demonstrated as effective in Table 1, and pointed out in L344-346 in the main paper.
>
> 3. Intuitively, **binary latent codes** provide more compact feature space, which can better discriminate visual features. Many works have studied this topic and demonstrate supportive evidence [1,2,3,4].
>
> ### W2: Improvement with Bernoulli diffusion process and binary codes
>
> The improvement of overall results comes from two perspectives: **representation capabilities** which have been discussed above, and **generation performance**.
>
> Regarding generation performance, the Bernoulli diffusion process was first proposed in [5] for image generation, demonstrating that it is one of the most effective methods for modeling binary codes.
> Using binary codes provides a distinct advantage over encoding images as continuous values, as in VAE, or as discrete indices, as in VQVAE. Traditional image autoencoders face longstanding issues, such as "*posterior collapse*" in VAE and "*low codebook utilization*" in VQVAE.
> Binary autoencoders (B-AE) eliminates the need for codebooks, offering a compact yet expressive representation of images in a binary latent space. Our model is built upon these binary codes.
>
> We provide a reconstruction FID (rFID) comparison between B-AE and VQVAE (used in LlamaGen) in our **response to Reviewer FMho (W2)**. B-AE with code dimensions greater than 20 achieves lower rFID than VQVAE, which highlights the advantages of B-AE.
>
> [1] Cakir et al. Hashing with mutual information. TPAMI 2019.
> [2] Jiang et al. Asymmetric deep supervised hashing. AAAI 2018.
> [3] Wei et al. A^2-NET: Learning attribute-aware hash codes for large-scale fine-grained image retrieval. NeurIPS 2021.
> [4] Wu et al. Deep incremental hashing network for efficient image retrieval. CVPR 2019.
> [5] Wang et al. Binary Latent Diffusion. CVPR 2023.

---

> ### Author Response · Authors · 2024-11-19
> **Response to Reviewer nzvc (part 2/2)**
>
> ### W3: More explanation of bidirectional attention
>
> Technically, replacing causal attention with bidirectional attention is straightforward: **simply use all-one masks instead of causal masks.**
>
> ***No fine-tuning involved.*** We would like to clarify that we use bidirectional attention, and train BiGR with masked modeling **from scratch**. Therefore, there is *no* fine-tuning stage in training. During training, we simply compute losses for masked positions. The detailed description of how we train our model can be found in Sec. 3.2 in the main paper.
>
> ***Inference.*** There are two inference scenarios: generation and representation extraction.
>
> 1. For **generation**, we design entropy-ordered sampling to iteratively unmask tokens from a full mask sequence, as detailed in Sec. 3.3 in the main paper.
> 2. For **representation extraction**, we input the full image into the model without any masks and use the intermediate features as the image representation. Please see details in L251-257.
>
> Both inference processes work smoothly with bidirectional attention.
>
> Note that, in both training and inference, the model uses bidirectional attention. Therefore, *no* special modifications are needed when switching between stages.
>
> ### W4: Results in LlamaGen paper
>
> We report the results from Table 8 in the Appendix of the LlamaGen paper. These results are conducted under a **strict 256$\times$256 resolution setting**, which is a *fair* comparison to our setting and is commonly used by most related papers.
>
> The results in Table 6 of LlamaGen's main paper, which the reviewer may refer to, are based on a setting where the generated images have a resolution of 384$\times$384 and are resized to 256$\times$256 for metric evaluation. LlamaGen does not emphasize this detail in their main paper.
> More detailed results for the same setting are provided in Tables 9 and 10 in the Appendix of the LlamaGen paper. The results match those reported in Table 6. However, this setting is *not* a fair comparison, as our model directly generates 256$\times$256 images without resizing.
>
> We kindly encourage the reviewer to cross-check **Tables 6, 8, 9 and 10** in the LlamaGen paper.
>
> Thus, the LlamaGen's results reported in our paper provide a **fair comparison to ours**. We clarified this point in L322-323 of the revised paper.
>
> ### Q1: SiT
>
> Thank you for pointing out this great work, SiT!
>
> We added this citation at L53 and L140. We included the SiT results in Table 5 and Table 9, and highlighted the state-of-the-art results achieved by SiT among diffusion-based models in L467-468. All changes are reflected in the revised paper.
>
> ### Q2: Citation format error
>
> Thanks! We fixed the citation format at L196.
>
> ---
>
> We deeply appreciate the reviewer's efforts in helping make our paper more clear and stronger.
>
> **If there are any further questions, we are happy to dicuss them!**

---

> > ### Author Response · Authors · 2024-11-21
> > **Follow-up**
> >
> > Could you please confirm if our response has addressed your concerns? Feel free to ask any questions or discuss further!
> >
> > Thank you so much!

---

> > > ### Comment · Reviewer_nzvc · 2024-11-25
> > > **Thanks for the Response**
> > >
> > > I find my concerns to be addressed in the rebuttal, and I am happy to increase my score.

---

> > > > ### Author Response · Authors · 2024-11-25
> > > > **Thank you!**
> > > >
> > > > Thank you so much! We are very glad that your concerns have been addressed!

---

### Author Response · Authors · 2024-11-19
**Global response and summary of changes in the revision**

We thank all reviewers for their time and effort in reviewing our paper!

Below, we summarize the changes made in the revised paper:

1. We clarified the fair comparison setting between LlamaGen and BiGR in L322-323. (`nzvc`)
2. We added the citation of SiT at L53 and L140, the SiT results in Tables 5 and 9, and the claim highlighting the state-of-the-art results achieved by SiT among diffusion-based models in L467-468. (`nzvc`)
3. We fixed the citation format at L196. (`nzvc`)
4. We added the observations of the inference hyperparameters in L707-718 in Appendix A. (`jehB`,`bv3t`)
5. We added the clarification that the binary transcoder component is responsible for denoising in L207-208 of Bernoulli diffusion paragraph. (`jehB`)
6. We visualized the generated results at different iterations in entropy-ordered sampling in Appendix C. (`jehB`)
7. We added failure cases in Appendix D. (`jehB`)
8. We added **text-to-image generation results** using BiGR in Appendix B. (`FMho`)
The generated images are shown in Figure 7 (with short prompts) and Figure 8 (with long prompts). **We kindly encourage all reviewers to take a look!**

The revised content is highlighted in purple.

We sincerely thank all reviewers again for their valuable suggestions, which have greatly helped strengthen our paper.

If you have any further questions, we would be happy to discuss them!

---

### Meta-Review · Area_Chair_CZSB · 2024-12-24

**Metareview:**

This paper introduces BiGR, a conditional image generation model that leverages compact binary latent codes. Unlike previous masked autoregressive approaches, BiGR employs a binary tokenizer and utilizes Bernoulli diffusion for binary code generation. These two key design choices significantly improve the model's performance. The authors validate BiGR's generative quality and representation capabilities through extensive experiments and demonstrate its applicability in applications such as inpainting, outpainting, and zero-shot generalization. Given the positive feedback from all reviewers, I recommend the acceptance of this paper.

**Additional Comments On Reviewer Discussion:**

The reviewers raised questions about the underlying intuition behind the performance improvements achieved by the binary tokenizer and Bernoulli diffusion. The authors provided detailed responses to address these concerns.

---

### Decision · Program_Chairs · 2025-01-22

Accept (Poster)